# Technical Note: Use of an Atmospheric Simulation Chamber to Investigate the Effect of Different Engine Conditions on Unregulated VOC-IVOC Diesel Exhaust Emissions

Kelly L Pereira[1], Rachel Dunmore[1], James Whitehead[2], M. Rami Alfarra[2,3], James D. Allan[2,3], Mohammed S. Alam[4], Roy M. Harrison[4,5], Gordon. McFiggans[2], Jacqueline F. Hamilton[1].

[1]Wolfson Atmospheric Chemistry Laboratories, Department of Chemistry, University of York, York, YO10 5DD, UK
[2]School of Earth, Atmospheric and Environmental Sciences, University of Manchester M13 9PL, UK
[3]National Centre for Atmospheric Science, UK
[4]School of Geography, Earth and Environmental Sciences, University of Birmingham B15 2TT, UK
[5]Department of Environmental Sciences / Center of Excellence in Environmental Studies, King Abdulaziz University, PO Box 80203, Jeddah, 21589, Saudi Arabia

*Correspondence to*: Jacqueline Hamilton (jacqui.hamilton@york.ac.uk)

## Abstract

Diesel exhaust emissions were introduced into an atmospheric simulation chamber and measured using thermal desorption comprehensive two-dimensional gas chromatography coupled to a flame ionisation detector (TD-GC×GC-FID). An extensive set of measurements were performed to investigate the effect of different engine conditions (*i.e.* load, speed, 'driving scenarios') and emission control devices (with/without diesel oxidative catalyst, DOC) on the composition and abundance of unregulated exhaust gas emissions from a light-duty diesel engine, fuelled with ultra-low sulphur diesel (ULSD). A range of exhaust dilution ratios were investigated (range = 1:60 to 1:1158), simulating the chemical and physical transformations of the exhaust gas from near to downwind of an emission source. In total, 16 individual and 8 groups of compounds (aliphatics and single-ring aromatics) were measured in the exhaust gas ranging from volatile to intermediate volatility (VOC-IVOC), providing both detailed chemical speciation and groupings of compounds based on their structure and functionality. Measured VOC-IVOC emission rates displayed excellent reproducibility from replicate experiments using similar exhaust dilution ratios. However, at the extremes of the investigated exhaust dilution ratios (comparison of 1:60 and 1:1158), measured VOC-IVOC emission rates displayed some disagreement owing to poor reproducibility and highlighted the importance replicate sample measurements. The investigated DOC was found to remove $43 \pm 10$ % (arithmetic mean $\pm$ experimental uncertainty) of the total speciated VOC-IVOC ($\sum$SpVOC-IVOC) emissions. The compound class dependant removal efficiencies for the investigated DOC were $39 \pm 12$ % and $83 \pm 3$ % for the aliphatics and single-ring aromatics, respectively. The DOC aliphatic removal efficiency generally decreased with increasing carbon chain length. The $\sum$SpVOC-IVOC emission rates varied significantly with different engine conditions, ranging from 70 to 9268 mg kg$^{-1}$ (milligrams of mass emitted per kilogram of fuel burnt). $\sum$SpVOC-IVOC emission rates generally decreased with increasing engine load and temperature, and to a lesser

degree, engine speed. The exhaust gas composition changed considerably as a result of two influencing factors, engine combustion and DOC hydrocarbon (HC) removal efficiency. Increased engine combustion efficiency resulted in a greater percentage contribution of the $C_7$ to $C_{12}$ n-alkanes to the $\sum$SpVOC-IVOC emission rate. Conversely, increased DOC HC removal efficiency resulted in a greater percentage contribution of the $C_7$ to $C_{12}$ branched aliphatics to the $\sum$SpVOC-IVOC emission rate. At low engine temperatures (< 150°C, below the working temperature of the DOC), the contribution of n-alkanes in the exhaust gas increased with increasing combustion efficiency and may be important in urban environments, as n-alkanes are more efficient at producing SOA than their branched counterparts. At very high engine temperatures (maximum applied engine speed and load, engine temperature = 700°C), the n-alkane contribution increased by a factor of 1.6 times greater than observed in the cold-start experiment (most similar to unburnt fuel) and may suggest liquid fuel based estimates of secondary organic aerosol (SOA) yields may be inconsistent with exhaust SOA yields, particularly at high engine speeds and loads (*i.e.* high engine temperatures). Emission rates were found to be 65 times greater from a cold-start experiment than at maximum applied engine speed and load. To our knowledge, this is the first study which uses an atmospheric simulation chamber to separate the effects of the DOC and combustion efficiency on the exhaust gas composition.

## 1. Introduction

Urban air pollution is detrimental to human health, adversely effecting air quality and resulting in increased morbidity and mortality rates (Han and Naeher, 2006; Cohen et al., 2005; Prüss-Üstün and Corvalán, 2006). The World Health Organisation attributed 1.34 million premature deaths to urban air pollution in 2008 (WHO, 2006; Krzyzanowski and Cohen, 2008). Of these deaths, 1.09 million could have been prevented if the air quality guidelines had been met (WHO, 2006; Krzyzanowski and Cohen, 2008). Over half of the world's population now live in urban areas (Prüss-Üstün and Corvalán, 2006; UnitedNations, 2014). By 2050, this population is expected to grow to 6.34 billion people, with an estimated 66% of the world's population living in urban environments (Prüss-Üstün and Corvalán, 2006; UnitedNations, 2014). Road transport emissions are a dominant source of urban air pollution (DEFA, 1993; Colvile et al., 2001; HEI, 2010) with common road-traffic pollutants including gaseous hydrocarbons (including volatile organic compounds, VOCs), nitrogen oxides (sum of NO + $NO_2$), carbon oxides (CO and $CO_2$) and particulate matter (PM), with secondary reaction processes resulting in the formation of ozone and secondary aerosol (WHO, 2006; HEI, 2010). Exposure to road-traffic air pollutants, both primary and secondary, are of a major health concern (UnitedNations, 2014; WHO, 2006; HEI, 2010). Secondary aerosol formation from diesel and gasoline powered motor vehicles has received considerable attention in recent years (Gentner et al., 2017). There is currently considerable debate as to whether diesel or gasoline powered motor vehicles are more important for secondary organic aerosol (SOA) formation and which precursors are the most efficient at forming SOA (Gentner et al., 2017). In Europe, almost half of all new passenger cars are diesel (49.5%), with petrol (45.8%), electric hybrids (2.1%), electric (1.5%) and alternative fuels (1.2%) accounting for the remaining fraction (ACEA, 2016). Diesel exhaust emissions vary considerably with vehicle type, age, operation conditions, fuel, lubricant oil and emission control devices, among other factors (HEI, 2010). Emission

regulations of nitrogen oxides, carbon monoxide, PM and total hydrocarbon mass has resulted in the reduction of exhaust emissions (HEI, 2010). However, this 'blanket approach' for the reduction of total hydrocarbon mass, has in-part, resulted in few studies investigating the detailed chemical composition of exhaust emissions with varying engine conditions (Yamada et al., 2011). Another contributing factor, is the difficulty in exhaust gas measurement (Yamada et al., 2011; Rashid et al., 2013).

On-road measurements of exhaust gas are difficult, due to the continually evolving chemical composition, requiring techniques capable of providing detailed chemical speciation in real-time, or near-real-time. Furthermore, the vast number of gaseous compounds in exhaust emissions often involves lengthy quantification processes. The detailed chemical characterisation of exhaust gas with varying engine conditions however, can considerably aid emission inventories and provide a greater understanding of exhaust emissions on local air quality. In addition, this information could serve to influence the design of

emission control devices, reducing the emission rates of potentially harmful unregulated exhaust gas components.

On-road measurements of unregulated exhaust gas emissions are often performed in tunnels, on roadsides, or motorways (*e.g.* (Gentner et al., 2013; Liu et al., 2015; Ježek et al., 2015; Zavala et al., 2006; Jiang et al., 2005; Kristensson et al., 2004; Fraser et al., 1998; Miguel et al., 1998; Staehelin et al., 1998)). These measurements provide a compositional overview of the exhaust

emissions from the on-road vehicular fleet, consisting of a vast range of vehicle types (*e.g.* light-duty, heavy-duty), emission control devices (*e.g.* with/without exhaust gas recirculation) and fuel composition (*e.g.* ULSD, super unleaded petrol, premium unleaded petrol, biofuel, among others). These measurements however, do not allow the effect of different engine conditions or emission control devices on the exhaust gas composition to be investigated. Dynamometer engines or chassis dynamometers can afford compositional insight into exhaust emissions with varying engine conditions, providing a high degree of control

and reproducibility (Tadano et al., 2014; Louis et al., 2016). Several studies have used dynamometers to investigate compositional changes in unregulated exhaust gas emissions with the use of different transient driving cycles, for petrol engines (Pang et al., 2014; Baldauf et al., 2005), diesel (Yamada et al., 2011; Cross et al., 2015; Schauer et al., 1999; Zhao et al., 2015; Ballesteros et al., 2014; Nelson et al., 2008; Siegl et al., 1999; Westerholm et al., 1991), or both (Alves et al., 2015; Chirico et al., 2014; Alkurdi et al., 2013; Caplain et al., 2006; Schmitz et al., 2000; Louis et al., 2016). Driving cycles (often performed

with a chassis dynamometers) are designed to simulate real-world driving conditions, allowing the exhaust emissions from individual vehicles to be investigated. However, these driving cycles offer limited information on the effect of combustion or specific engine conditions (*e.g.* engine load, speed) on unregulated exhaust emissions, due to the averaging of emissions over entire driving cycles and lack of steady-state engine conditions (Cross et al., 2015); compositional information which is easily obtained with the use of a single-engine dynamometer rig (Chin et al., 2012; Cross et al., 2015; Zhu et al., 2013; Schulz et al.,

1999; Machado Corrêa and Arbilla, 2008). Recent studies have focused on the measurement of intermediate-VOCs in diesel exhaust emissions, primarily due to advances in instrumentation to allow the detection of these species. IVOCs have an effective saturation concentration ($C^*$) of $10^3$ to $10^6$ µg m$^{-3}$ and reside almost exclusively in the gas-phase at atmospheric conditions (Donahue et al., 2006). IVOCs comprise a considerable fraction of diesel exhaust emissions, with studies attributing ~ 20 to 60% of the non-methane organic gases to IVOCs (Schauer et al., 1999; Siegl et al., 1999; Zhao et al., 2015; Gordon et

al., 2014). IVOC diesel exhaust emissions are relatively poorly characterised, yet contribute significantly to SOA formation (Gentner et al., 2017). Recently, Cross et al. (2015) investigated IVOC diesel exhaust emissions using a dynamometer rig. It was found that IVOC diesel exhaust emissions were highly dependent on engine power. At low engine loads, the exhaust gas composition was dominated by saturated hydrocarbons from unburnt fuel. At high engine loads however, the exhaust gas

composition changed, including newly formed unsaturated hydrocarbons and oxidised compounds from incomplete combustion (Cross et al., 2015). Furthermore, Chin et al. (2012) found the composition of VOC and IVOC diesel exhaust emissions (generated under steady-state engine conditions) depended on engine load, fuel type (ULSD and biodiesel) and emission control devices.

This study investigates the compositional changes of unregulated exhaust emissions with varying engine conditions (*i.e.* engine load, speed and 'driving scenarios') and emission control devices (with/without DOC) using a dynamometer rig. In contrast to previous studies, this work combines both detailed chemical speciation and groupings of VOC–IVOCs based on their structure and functionality, providing a more detailed compositional overview of the effect of different engine conditions on exhaust gas emissions. This work also investigates the effect of different exhaust dilution ratios on the exhaust gas composition,

simulating the chemical and physical transformations of the exhaust gas emissions at varying ambient dilutions (*i.e.* near to downwind of an emission source). The emissions from a light-duty 1.9L Volkswagen diesel engine were investigated. Exhaust emissions from different engine conditions were introduced into an atmospheric chamber which was used as a 'holding-cell' for sampling, allowing lower time resolution techniques to be used. In total, 16 individual and 8 groups of compounds were measured in the exhaust gas using thermal desorption comprehensive two-dimensional gas chromatography coupled to a flame

ionisation detector (TD-GC×GC-FID). The effect of different engine conditions and emission control devices on the composition and abundance of the speciated VOC-IVOCs is discussed, along with the reproducibility of the engine exhaust emissions/chamber sampling system and the potential impacts of our findings.

## 2. Experiment design

### 2.1 Experiments

A series of experiments were performed in July to August 2014, November 2014, and September to October 2015, as a part of the project, COMbustion PARTicles in the atmosphere (Com-Part). Experiments were designed to systematically characterise the chemical and physical transformations of primary and secondary particles emitted from a light-duty diesel engine under a range of atmospheric dilution and oxidation conditions. The results shown here, focus on the effect of different engine conditions on the composition and abundance of VOC-IVOCs in the raw exhaust emissions, which formed a subset of the total

number of experiments performed. The experimental descriptions and engine operating parameters discussed here, can be found in Table 1. A range of engine conditions were studied, including: (i) engine speed, ranging from 1150 rpm (idle) to 3000 rpm (maximum engine output), (ii) engine load, ranging from 0 % (no load) to 53 % (the maximum load which could be safely

applied to by the dynamometer in the experimental setup), (iii) emission control devices (with and without the DOC) and, (iv) 'driving scenarios' (see below for further details). Exhaust dilution ratios were varied to represent a range of ambient conditions from near to downwind of an emission source, capturing the chemical and physical transformations of semi-volatiles in the exhaust emissions with varying ambient dilutions.

The majority of experiments focused on steady-state engine conditions, where selected engine running parameters were applied to the engine and the engine temperature allowed to stabilise, prior to the introduction of the exhaust emissions into the Manchester Aerosol Chamber (MAC). The MAC was filled with clean air prior to the introduction of the exhaust emissions. Steady-state engine conditions are defined here, as a constant engine temperature within ±10% of the steady-state average. Steady-state engine conditions were not performed for the cold-start (exp. 6, 7 and 14, Table 1) and 'driving scenario' (exp. 8 and 9, see Table 1) experiments, with the cold-start experiments requiring exhaust injection into the MAC after ~ 1 to 2 minutes of engine start-up (*i.e.* cold-engine). The driving scenario experiments involved a sequence of engine conditions, with the injection of the exhaust emissions into the MAC after the completion of the sequence, but prior to achieving steady-state engine conditions. The sequence of engine conditions used in the driving scenario experiments are discussed in section 3.3 and shown in the SI Figure S1. All experiments except experiment 3 (see Table 1) were performed with the DOC. This allowed the combined effect of the DOC and different engine conditions on the exhaust emissions to be observed, *i.e.* engine conditions most representative of on-road diesel vehicles.

## 2.2 Chamber setup

Experiments were performed in the MAC located within the University of Manchester, UK. The MAC consists of an 18 m$^3$ fluorinated ethylene propylene Teflon bag with the following dimensions; 3m (L) × 3m (W) × 2m (H). The chamber is supported by three rectangular aluminium frames, two of which are free moving, allowing the chamber to expand and collapse as sample air flow is introduced or extracted. Purified air is used within the chamber and is humidified prior to introduction. A suite of instruments was used to measure chamber temperature (series of cross-calibrated thermocouples), relative humidity (Dewmaster chilled mirror hygrometer, Edgetech Instruments, USA), $CO_2$ (model 6262, Li-Cor Biosciences, USA), $NO_x$ (model 42i, Thermo Scientific, MA, USA), $O_3$, (model 49C, Thermo Scientific, MA, USA) and VOC-IVOCs (GC×GC-FID, see below for further details). Particle number, mass and diameter were measured using a differential mobility particle sizer (DMPS (Williams et al., 2007)) consisting of a differential mobility analyser (DMA (Winklmayr et al., 1991)) and a condensation particle counter (CPC, model 3010, TSI Inc., USA). Further technical information regarding the chamber design can be found in Alfarra et al. (2012).

## 2.3 Engine and exhaust sampling system

The emissions from a light-duty Volkswagen (VW) 1.9L diesel engine were investigated. A schematic of the dynamometer and exhaust sampling system is shown in Figure 1. The engine had 4 cylinders with a capacity of 1896 cm$^3$ and a compression

ratio of 19.5:1. The engine was mounted on an eddy current dynamometer rig (CM12, Armfield Ltd, Hampshire, UK) and the exhaust connected to a new (0 mileage hours) retrofitted DOC. The DOC was purchased from a local garage (Oldham Tyre and Exhaust, Oldham, UK) and consisted of a mix of platinum and rhodium. No diesel particulate filter was used, conforming to Euro 4 emission control regulations. The auto-equivalent version of this engine has been used in several VW polo and Jetta

models in the early 2000's and was chosen as an example of light-duty diesel engine. The aftertreatment was selected to meet Euro 4 emission control regulations required for such models. Engine running parameters (*i.e.* speed, load, and throttle) were controlled using a dedicated software package (Armfield Ltd, Hampshire, UK) on a separate PC. Engine load is the amount of breaking force (mechanical resistance) applied to engine during operation, simulating weight (*i.e.* load) and/or resistance. Engine throttle controls the amount of fuel injected into the combustion chamber required for speed (rate of movement,

measured in rpm) and load. Engine torque (force generated in the crankshaft of the engine) was not controlled but is included in Table 1 for reference. The engine temperature was measured *via* an in-built thermocouple located inside the engine exhaust pipe, next to the engine. The DOC temperature has been inferred from the measured exhaust temperature. The DOC temperature will be lower than the measured exhaust temperature due to the DOC being located further down the exhaust pipe. A 2-meter-long, 2-inch bore, stainless steel tube with a computer controlled pneumatic valve, was used to allow the engine

emissions to be introduced into the MAC or diverted to waste. The timed control of the pneumatic valve allowed a proportion of the exhaust emissions to be introduced into the chamber, controlling dilution. The final exhaust dilution ratios were calculated from the measured $CO_2$ concentration prior to and after the introduction of the exhaust emissions. The exhaust dilution calculations will be discussed in further detail in a separate publication. The engine was fuelled with standard European (EN590 specifications, Euro 5 compliant) ULSD obtained from a local fueling station. Two batches of fuel were obtained, the

first in June 2014 (batch A) and the second in November 2014 (batch B). A second batch of fuel was required due to a considerable increase in the number of planned experiments. The fuel batches were of the same specification and obtained from the same local fueling station. Batch A was used in experiments 1 to 9 and batch B in experiments 10 to 16 (see Table 1 and section 3.1.1). The sulphur content was < 10 ppm. Further information regarding the standard European ULSD fuel specifications can be found in 2009 EC directive (EU, 2009).

**2.4 TD-GC×GC-FID**

VOC-IVOC exhaust emissions were measured using thermal desorption comprehensive two-dimensional gas chromatography with a flame ionisation detector (TD-GC×GC-FID) operating at 200 Hz. A TT24-7 thermal desorption unit (Markes International, Llantrisant, UK) with an air server attachment was used for sample collection. The inlet of the TD unit was connected to MAC using ~ 2.5 meters of heated 1/4" stainless steel tubing. The stainless steel tubing was heated to ~ 70 °C to

reduce condensational losses of VOCs. An in-line unheated particulate filter prevented sampled particles from entering the TD unit. The in-line filter was replaced prior to each experiment to minimise particulate loadings. A clean air diaphragm pump (model PM25602-86, KNF Neuberger, Oxfordshire, UK) was used to extract an overflow of sample air from the MAC, a proportion of which was sampled into the TD unit. Two sequential glass traps cooled to -20 ˚C in an ethylene glycol bath were

used to remove water vapour from the sampled air. No significant VOC losses have been found using this method of water vapour removal (Lidster, 2012). Air samples were trapped onto Tenax sorbent tubes (Markes International, Llantrisant, UK) held at -10°C during sampling (26 minute sampling duration) and heated to 230°C upon desorption.

An Agilent 7890 GC (Agilent Technologies, Wilmington, USA) with a modified modulation valve, consisting of a 6-port, 2-way diaphragm valve (Valco Instruments, Texas, USA) and 50 µL sample loop (Thames Resteck, UK) was used (see Lidster et al. (2011) for further information). Cryogenic cooling (liquid $CO_2$, BOC, UK) was used to re-focus the sample on the head of the primary column upon desorption. Compound separation was achieved using a primary 25 meter 5% phenyl polysilphenylene-siloxane column (BPX5, SGE, Ringwood, Australia) with a 0.15 mm internal diameter and 0.4 µm film thickness, and a secondary 7-meter polyethylene glycol column (BP20 SGE, Ringwood, Australia) with a 0.25 mm internal diameter and 0.25 µm film thickness. Helium (CP grade, BOC, UK) was used as the carrier gas. Primary and secondary column pressures were controlled using an electronic pneumatic control (Agilent 7890 EPC) and were set at 50 and 23 psi, respectively. The modulator was heated to 120°C with a 5 second cycle time, comprising of 0.3 second injection and a 4.7 second sample introduction. The oven temperature program consisted of a two-stage ramp; holding at 70°C for 1 minute, increasing to 160°C at 16 minutes (6°C min$^{-1}$), then 200°C at 20 minutes (10°C min$^{-1}$) with an additional 2-minute hold, giving a total runtime of 22 minutes. The FID heater was set to 300°C with a hydrogen flow of 30 ml min$^{-1}$ (CP grade, BOC, UK) and an air flow of 300 ml min$^{-1}$ (BTCA 178 grade, BOC UK).

A National Physical Laboratory (NPL30, Teddington, UK) gas standard was used to monitor instrument variability over the course of the experiments. VOC-IVOC concentrations were determined using either the NPL gas standard or the relative response factors (RRF) of liquid standards (see SI for further information). Calibrations were performed weekly using the NPL gas standard, or more frequently during instrument maintenance periods. The instrument detection limits for the investigated compounds can be found in the SI of Dunmore et al. (2015). Compounds with carbon numbers greater than $C_{15}$ were not measured due to instrument temperature constraints; the boiling points of these compounds are too high to be removed from the column at the maximum operating temperature of the modulator. The exhaust injection times into the MAC, GC×GC-FID exhaust sampling start and end times, and the number of replicate measurements performed in each experiment, is shown in SI Table S2. Only samples where no changes had been made to the chamber conditions were analysed. The exhaust emissions were blank subtracted using the chamber background measurement/s prior to the introduction of the exhaust emissions. The average sampling start time was 13 minutes after the injection of the exhaust emissions into the MAC. No apparent losses of the VOC-IVOCs were observed during sampling; the average relative standard deviation from replicate measurements of the investigated VOC-IVOCs over the longest sampling duration (~ 2 hours) was 6.4% (exp. 6, see Table 1 and SI Table S2).

**2.5 Liquid fuel analysis**

The two batches of ULSD fuel (see section 2.3 and 3.1.1) were analysed using comprehensive two-dimensional gas chromatography (model 6890N, Agilent Technologies, UK) coupled to a time-of-flight mass spectrometer (Pegasus 4D, Leco, MI, USA) (GC×GC-TOFMS). Compound separation was achieved using a primary 15 meter 5% phenyl polysilphenylene-

siloxane column (BPX5, SGE, Ringwood, Australia) with a 0.25 mm film thickness and 0.25 mm internal diameter, and a secondary 2 meter 50% phenyl polysilphenylene-siloxane column (BPX50, SGE, Ringwood, Australia) with a 0.25 mm film thickness and 0.25 mm internal diameter. Neat ULSD fuel was introduced into the GC×GC-TOFMS using a Gerstel multipurpose sampler (MPS 2, Gerstel, USA) with dedicated controller (model C506, Gerstel, USA). A 1 µL injection volume was used with a split ratio of 100:1. The transfer line was set to 270°C. Cryo-jet modulation cooling was used to achieve

comprehensive two-dimensional separation. Helium (CP grade, BOC, UK) was used as the carrier gas with a constant flow rate of 1.5 ml min$^{-1}$. The oven starting temperature was set to 65°C with a 0.2 minute hold, followed by a temperature ramp of 4°C min$^{-1}$ to 240°C, with a further 10 minute hold. The modulator and secondary oven temperature was set to 15°C and 20°C above the oven temperature, respectively. The TOFMS acquisition rate was set to 50 spectra per second, with a scan range of mass-to-charge (*m/z*) 35 to 500. The data was analysed using Leco ChromaTOF software version 4.51.6 (Leco, MI, USA).

Compounds were identified using the National Institute of Standard and Technology (NIST) standard reference database (version 11).

**3. Results and discussion**

The effect of different engine loads, speeds, 'driving scenarios' and emission control devices (*i.e.* with/without DOC) on the VOC-IVOC emission rates from a light-duty diesel engine fuelled with USLD is discussed. In total, 16 individual and 8 groups

of VOC-IVOCs were speciated in the exhaust gas using TD-GC×GC-FID. The individual compounds included nine single-ring aromatics: benzene, toluene, ethyl benzene, meta- and para-xylene (grouped), ortho-xylene, styrene, 1,3,5-TMB, 1,2,4-TMB and 1,2,3-TMB, and 7 *n*-alkanes from *n*-heptane to *n*-tridecane. Grouped compounds consisted of $C_7$ to $C_{13}$ branched aliphatics grouped by carbon number and single-ring aromatics with three carbon substitutions (*i.e.* those in addition to the trimethylbenzene isomers above). The emission rates of *n*-tridecane and the $C_{13}$ branched aliphatic grouping were not measured

in some experiments due to a shift in the instrument retention time, resulting in these species not being observed. The saturation concentration ($C^*$, µg m$^{-3}$ (Donahue et al., 2006)) of the speciated compounds ranged between $10^5$ –$10^8$ µg m$^{-3}$, classifying these species as intermediate to volatile organic compounds (VOC-IVOCs). The most abundant volatility fraction of IVOCs from diesel exhaust emissions were measured, see Zhao et al. (2015) for further information. An annotated chromatogram displaying the speciated compounds is shown in Figure 2. The use of two different stationary phases in GC×GC allows

compounds to be separated by two physical properties, such as boiling point and polarity, as shown here (see Figure 2). This two-dimensional separation creates a characteristic space where compounds are grouped by similar physical properties (*e.g.* aromatic and aliphatic bandings (see Figure 2, *c.f.* Hamilton and Lewis (2003), Dunmore et al. (2015)), aiding in the

identification of unknowns. This characteristic space, in combination with the use of commercially available standards and the elution patterns observed in previous work using this instrument (Dunmore et al., 2015), allowed 8 compound groupings to be identified. The identification of all the individual compounds (except styrene, see SI) were confirmed using commercially available standards. The emission rates of the individual and grouped compounds and their percentage contribution to the total

speciated VOC-IVOC emission rate (hereafter referred to as ∑SpVOC-IVOC), are shown in the SI Tables S3 to S6. No corrections have been made for gas-phase absorption to PM in this work. Gas-phase absorption to PM is negligible due to the relatively high vapour pressures of the compounds speciated, low VOC-IVOC mixing ratios and small amount of aerosol mass present after exhaust dilution.

## 3.1 Experimental reproducibility

The reproducibility of the measured VOC-IVOC emission rates with different engine conditions and exhaust dilution ratios were investigated and are discussed below. The emission rates from two replicate cold-start experiments (1150 rpm, 0% load, exp. 6 and 7, see Table 1) and two replicate warm with load (WWL) experiments (2000 rpm, 30% load, exp. 15 and 16, see Table 1) are shown in Figures 3 and 4, respectively. Both replicate experiments were performed with similar exhaust dilution ratios. The emission rates from the replicate cold-start and WWL experiments displayed excellent reproducibility, considering

the vast number of variables in these experiments (*e.g.* combustion and DOC HC removal efficiency). All emission rates, except styrene in one experiment (< limit of detection), were observed to be within the calculated uncertainty (see SI). The emission rates from two replicate warm high load experiments (2500 rpm, 40% load, exp. 1 and 2, see Table 1) with different exhaust dilution ratios is shown in Figure 5. The exhaust dilution ratios in these experiments were 166 and 313 in experiments 1 and 2 (see Table 1), respectively. The emission rates in these experiments are relatively comparable. Only one measurement

of the exhaust emissions was performed in each experiment. The majority of experiments had a minimum of two replicate measurements of the exhaust emissions (see SI Table S2), possibly accounting for slight differences observed in the measured VOC-IVOC emissions rates.

The emission rates from two replicate WWL experiments (2000 rpm, 30% load, exp. 4 and 5, see Table 1) at the extremes of

the investigated exhaust dilution ratios are shown in Figure 6. The exhaust dilution ratios were 1158 and 60 in experiments 4 and 5, respectively. The emission rates in these experiments displayed some disagreement. The engine thermocouple was unresponsive during one of these experiments (exp. 4, see Table 1). Consequently, it is not known if steady-state engine conditions were achieved prior to the introduction of the exhaust emissions into the MAC and whether the engine temperature upon injection was comparable to the replicate experiment, possibly accounting for the observed differences in the VOC-IVOC

emission rates. Nevertheless, no experiments with such large differences in the exhaust dilution ratios have been directly compared in the following work and where engine conditions are compared, experiments with similar exhaust dilution ratios and engine temperatures have been used. These experiments highlight the importance of replicate measurements and the comparison of VOC-IVOC emission rates from experiments with similar engine temperatures. A propagation of errors was

calculated to determine the experimental uncertainty in the measured emission rates and is discussed in detail in the SI. Briefly, the experimental uncertainty includes, (i) the standard deviation in the replicate measurements of the calibration standard and the reported uncertainty in the standard mixing ratios (where applicable), (ii) a 5% standard deviation in the chamber volume, and (iii) an additional 20% error in the emission rates of compounds integrated using the GC image software package, see the SI for further information. The experimental uncertainty in the measured emission rates for the investigated VOC-IVOCs, ranged from 6 to 50 % with an average of 22%.

### 3.1.1 ULSD fuel: Batch A and B

Two batches of ULSD fuel were used in the experiments (see section 2.3). The emission rates from three replicate cold-start experiments, two using fuel batch A (exp. 6 and 7, see Table 1) and one using fuel batch B (exp. 14), are shown in Figure 7. From Figure 7, it can be observed that there is a considerable difference in the emission rates of the $C_7$ to $C_{12}$ branched aliphatics between replicate experiments 6 and 7, and experiment 14. The emission rates of the $C_7$ to $C_{12}$ branched aliphatics decreased by a factor of ~ 4 with the use of fuel batch B (exp. 14). The excellent agreement of the emission rates between replicate cold-start experiments 6 and 7, suggests the compositional differences observed in experiment 14, is the result of a slight difference in the fuel composition between batches A and B. GC×GC-TOFMS was used to further investigate any compositional differences between the fuel batches. An extensive analysis of the diesel fuel was not performed. The aim of this analysis was to investigate whether there were any apparent differences in the fuel composition that would prevent a direct comparison of the emission rates from fuel batches A and B. An extracted ion chromatogram for $m/z$ 57 (dominant aliphatic fragment ion) from fuel batch A and B is shown in Figure 8A and B, respectively. The chromatograms were normalised to the total peak area to allow direct comparison of peak intensity between the chromatograms. The highlighted region in Figure 8 displays straight and branched chain aliphatics with a carbon number range of approximately $C_7$ to $C_{12}$ (determined from the NIST library). The peak intensities in the chromatograms from fuel batches A and B are largely comparable, except for the highlighted region, where a slightly lower peak intensity is observed in fuel batch B. As a result, the emission rates from fuel batches A and B have not been directly compared in the following work. The reason for the observed compositional differences between fuel batches A and B is unclear, although suggests a possible change in the refining process between the purchase of both fuel batches.

### 3.2 DOC removal efficiency

The HC removal efficiency of the DOC was investigated by performing two replicate experiments (exp. 2 and 3, Table 1) with and without the DOC. The additional backpressure created due to the in-line DOC appeared to have no effect on engine operation, allowing a direct comparison between both experiments. The engine speed and load were set to 2500 rpm and 40% load, respectively. The DOC HC removal efficiency is strongly dependant on working temperature. Below 200°C the DOC HC removal efficiency is close to zero, rising sharply to near 100% HC removal efficiency at ~ 430°C (Korin et al., 1999; Roberts et al., 2014; Majewski and Khair, 2006; Russell and Epling, 2011). The steady-state engine temperature in both

experiments was 450°C. Thus, the DOC was near maximum HC removal efficiency. The HC removal efficiency was calculated using the equation shown in Roberts et al. (2013). The removal efficiency of the investigated DOC for the speciated compounds is shown in Table 2. The DOC removed 43 ± 10 % (arithmetic mean ± experimental uncertainty, see section 3.1 and the SI for further information) of the ∑SpVOC-IVOC emissions. The compound class dependant removal efficiencies for the

investigated DOC were 39 ± 12 % and 83 ± 3 % for the aliphatics (branched and straight-chain) and single-ring aromatics, respectively. A typical DOC is expected to remove 50 to 70% of the total HC emissions (Johnson, 2001; Alam et al., 2016). For the investigated compounds, the total DOC HC removal efficiency is at the lower limit of this expectation. The DOC removal efficiency for styrene, m/p- and o-xylene, ethylbenzene ($C_2$ aromatic substitution grouping) and benzene was greater than 90%. In addition, the trimethylbenzenes (TMB) were not observed with the use of the DOC (~ 100% removal efficiency).

This high HC removal efficiency however, was not observed for all the single-ring aromatics. Toluene had a relatively poor removal efficiency in comparison, at 59 ± 9 %. Furthermore, the removal efficiency of the unspeciated $C_3$ aromatic substitution grouping (*i.e.* less branched aromatic isomers of TMB) was determined to be 63 ± 22 %, suggesting the isomeric structure influences removal efficiency, possibly the result of reactivity and/or adsorption to the metal binding sites in the DOC (*c.f.* (Salge et al., 2005; Russell and Epling, 2011)).

Generally, the HC removal efficiency decreased with increasing carbon chain length. This was particularly evident with the branched aliphatics, with the removal efficiency decreasing from 72% to 14 % from $C_7$ to $C_{12}$, with a sharp decrease in the removal efficiency from $C_{10}$ to $C_{12}$. Analogous to the branched aliphatics, the *n*-alkanes displayed the same rapid decrease in the HC removal efficiency between *n*-decane and *n*-dodecane, with the DOC observed to have no effect on the emission of *n*-

dodecane. The removal of *n*-alkanes in the DOC have been found to decrease with increasing carbon chain length, a result of the greater number of adjacent sites in the DOC required to achieve absorption (Yao, 1980; Russell and Epling, 2011), supporting the results shown here. However, recently Alam et al. (2016) investigated the HC removal efficiency of a similar specification DOC (*i.e.* mixed platinum and rhodium) for $C_{12}$ to $C_{33}$ *n*-alkanes, among other species. It was found that the DOC HC removal efficiency did not continue to decrease with increasing carbon chain length, rather decreasing from $C_{12}$ to $C_{16}$,

followed by an increase from $C_{17}$ to $C_{23}$ and further decrease from $C_{24}$ to $C_{32}$. Few studies have investigated the HC removal efficiency of individual species and grouped counterparts, expressing DOC HC removal efficiency as total HC, with no reference to possible compositional and structural effects, which based on the results shown in this work and Alam et al. (2016), require further study.

### 3.3 Driving scenarios

The VOC-IVOC emission rates from several driving 'scenarios' were investigated. The driving scenarios included either; (i) a single applied engine load and speed, and injection before a steady-state engine temperature had been achieved or, (ii) a sequence of different engine loads and speeds, during which steady-state engine temperature was achieved. These experiments were performed to gain a greater insight into the factors controlling VOC-IVOC emission rates. Three experiments were

performed, cold-start (exp. 6), cold loaded (exp. 8) and warm idle following load (WIFL, exp. 9). The engine conditions used in each of these experiments can be found in SI Figure S1. Cold-start included a cold-engine start (idling speed and 0% load) with the injection of the exhaust emissions into the MAC after ~ 1-2 minutes. Cold loaded included a cold engine start followed by the immediate application of 1500 rpm and 20% load, with a one minute hold before injection. Steady-steady engine temperatures were not achieved during the cold-start and cold loaded experiments. WIFL included a cold engine start, followed by the immediate application of 2000 rpm and 28-30% load with a 7-minute hold (during which a steady-state engine temperature was achieved), followed by one minute of idling speed (1150 rpm) and 0% load before injection. The $\sum$SpVOC-IVOC emission rates in each experiment were $9268 \pm 699$, $2902 \pm 199$ and $1438 \pm 96$ mg kg$^{-1}$ in the cold-start, cold loaded and WIFL, respectively. The application of 1500 rpm and 20% load for 1 minute (cold-loaded) resulted in a decrease in the $\sum$SpVOC-IVOC emissions by a factor of ~ 3, in comparison to the cold-start engine conditions; highlighting the importance of engine combustion efficiency on VOC-IVOC emission rates. The VOC-IVOC compositional profiles and emission rates in the driving scenario experiments can be observed in Figure 9.

The engine temperature in the cold-start and cold loaded experiments was 85°C and 169°C, respectively. In the WIFL experiment, the engine temperature reached 290°C during steady-state, decreasing to 150°C upon injection. The $\sum$SpVOC-IVOC emission rate was lower in the WIFL experiment than observed in the cold loaded experiment, where a higher engine temperature was measured upon injection. The HC removal efficiency of the DOC below 200°C is close to zero (Korin et al., 1999; Roberts et al., 2014; Majewski and Khair, 2006; Russell and Epling, 2011), suggesting the lower $\sum$SpVOC-IVOC emission rate observed in the WIFL experiment, is the result of increased combustion efficiency from the higher engine speed and load applied before idling conditions. Engine 'warm-up' increases the temperature of the lubricant, coolant and engine components, reducing friction and increasing combustion efficiency, thus resulting in less unburnt fuel emissions in the exhaust gas (*c.f.* (Roberts et al., 2014)). This increased combustion efficiency in the WIFL experiment is also supported by the exhaust gas composition (see Figure 9B). The engine temperatures in all three experiments were below 200°C and consequently, the DOC had a minimal effect on HC removal in these experiments. Therefore, the observed compositional changes in exhaust gas is the result of increasing combustion efficiency from the cold-start to WIFL experiment. Straight-chain alkanes are more easily fragmented during combustion than branched aliphatics (Fox and Whitesell, 2004). The sequential increase in the abundance of the $C_7$ to $C_{12}$ *n*-alkanes in the exhaust gas from the cold-start to WIFL experiment, suggests higher molecular weight *n*-alkanes which have not been measured ($> C_{13}$, abundant in diesel fuel and lubricate oil) undergo increasing fragmentation with increasing combustion efficiency, resulting in a higher percentage contribution of smaller *n*-alkanes (*i.e.* $C_7$ to $C_{12}$) to the $\sum$SpVOC-IVOC emission rates. The relationship between internal combustion efficiency and engine temperature is relatively linear (*e.g.*(Mikalsen and Roskilly, 2009)), with the exception of high engine loads and relatively low speeds (not performed here), where the engine combustion efficiency and temperature eventually plateau due to a too lean air/fuel ratio, resulting in incomplete combustion (see Haywood (1988) for further information). The percentage contribution of the $C_3$ aromatic substitution grouping to the $\sum$SpVOC-IVOC emission rates, displayed no obvious change with increasing

combustion efficiency (within the calculated uncertainty). However, the abundance of the $C_2$ aromatic substitution grouping and toluene generally decreased with increasing combustion efficiency, with the percentage contribution observed to plateau in the cold-start and cold loaded experiment, followed by a decrease in the WIFL experiment.

### 3.4 Engine load

The effect of different engine loads, at a constant speed, on the VOC-IVOC emission rates is discussed below. Three experiments were performed at 30, 40 and 53% engine load (exp. 12, 13 and 11, respectively, see Table 1). The GC×GC-FID was not operational during lower engine load experiments (not presented here). The $\sum$SpVOC-IVOC emissions were observed to decrease with increasing engine load, with $\sum$SpVOC-IVOC emission rates of 1019±65, 365±24 and 70±4 mg kg$^{-1}$ at 30, 40 and 53% load, respectively (see SI Table S7). This trend of decreasing VOC emission rates with increasing engine load has

been observed in a number of previous studies for light-duty and medium-duty diesel vehicles (Cross et al., 2015; Shirneshan, 2013; Chin et al., 2012; Yamada et al., 2011) and can be explained by considering the engine operation. At low engine temperatures (*i.e.* low engine loads and idling conditions), the fuel flow is increased to provide easily combustible conditions within the engine cylinder. This additional fuel flow creates a rich fuel/air ratio, where there is insufficient oxygen to burn the fuel, resulting in incomplete combustion and higher VOC-IVOC emission rates from the unburnt fuel. As the engine

temperature increases (*e.g.* with increasing engine load), the in-cylinder oxidation rate increases as the fuel components become more easily combustible at higher temperatures, increasing combustion efficiency and decreasing VOC-IVOC emission rates (Heywood, 1988). The effect of different engine loads, at a constant speed, on the VOC-IVOC emission rates is shown in Figure 10A. The carbon number distribution of the *n*-alkanes and branched aliphatics at 30% and 40% engine load are comparable. Branched aliphatics display an increase in abundance from $C_7$, reaching peak concentration at $C_{10}$, followed by a

decrease to $C_{13}$, similar to that observed in Bohac et al. (2006). Straight-chain alkanes do not display the same increase and decrease in abundance, with the emission rates of *n*-nonane and *n*-dodecane greater than *n*-undecane, displaying no obvious trend. At 53% engine load, the emission profile changes. The most abundant *n*-alkane and branched aliphatic grouping shifts to higher carbon numbers at higher loads, changing from *n*-nonane to *n*-undecane and from $C_{10}$ to $C_{12}$ branched aliphatics. The *n*-alkanes now display a sequential increase and decrease in their emission factors, as observed with the branched aliphatics.

This compositional shift to higher carbon number species under higher engine loads, has also been observed in Chin et al. (2012) for *n*-alkanes from an Isuzu 1.7L diesel engine fuelled with ULSD. Whilst no explanation was provided for this observation, Chin et al. (2012) found the most abundant *n*-alkane shifted from *n*-nonane at idling conditions (800 rpm) with no load, to *n*-tridecane at 2500 rpm with maximum applied engine load (900 brake mean effective pressure, kPa).

The percentage contribution of the individual and grouped VOC-IVOCs to the $\sum$SpVOC-IVOC emission rate in each experiment, is shown in Figure 10B. The percentage composition from a cold-start experiment (exp. 14) has also been included on the left of Figure 10B to provide a comparison between cold idle engine conditions (which has a compositional profile most similar to unburnt fuel) and different engine loads. The percentage contribution of the individual and grouped VOC-IVOCs to

the $\sum$SpVOC-IVOC emission rate changed considerably with different engine loads. All aromatics, except benzene, displayed a nonmonotonic behaviour with increasing engine load; their percentage contribution is high at cold idle and 40% load, with a smaller contribution at 30% and 53% load. This nonmonotonic behaviour has also been observed in Cross et al. (2015). Cross et al. (2015) investigated the load-dependant emissions from a 5.9L medium-duty diesel engine fuelled with ULSD. It was found that the fractional contribution of oxidised species and aromatics (not explicitly mentioned but shown in the data) varied inconsistently with increasing engine load. The reason for this nonmonotonic behaviour is currently unclear. The percentage contribution of benzene generally decreased with increasing engine load. Interestingly, the percentage contribution of the *n*-alkanes continued to decrease from cold idle to 40% load, followed by a considerable increase at 53% load. At 53% load, the *n*-alkanes represented 55% of the $\sum$SpVOC-IVOC emission rate, 1.6 times greater than observed in the cold-start experiment. Conversely, branched aliphatics displayed the opposite trend. The percentage contribution of the branched aliphatics continued to increase from cold idle to 40% load, followed by a considerable decrease at 53% load, to approximately the same percentage contribution observed in cold-start experiment.

This change in the percentage contribution of the *n*-alkanes and branched aliphatics at 53% engine load, can be explained by considering the DOC HC removal efficiency and the internal combustion temperature. From cold idle to 40% engine load, the engine temperature increased from < 100°C at cold idle to 445°C at 40% load. The DOC HC removal efficiency is thus increasing from near zero at cold idle to near maximum at 40% load. At 53% load, the steady-state engine temperature reached 700°C. The DOC HC removal efficiency was near maximum at 40% load and it is therefore unlikely that the DOC would account for such a considerable shift in the percentage contribution of the *n*-alkanes and branched aliphatics at 53% load. This shift in the composition is most likely the result of the considerably higher engine temperature, resulting in the fragmentation of higher molecular weight *n*-alkanes from increased internal combustion efficiency, as observed with the driving scenario experiments, resulting in a higher percentage contribution of $C_7$ to $C_{12}$ *n*-alkanes to the $\sum$SpVOC-IVOC emission rate at 53% load in comparison to cold idle. The compositional profiles from 0 to 40% engine load, display the combined effect of increasing engine combustion efficiency and DOC HC removal efficiency, possibly explaining why the percentage contribution of $C_7$ to $C_{12}$ *n*-alkanes do not increase with increasing engine combustion efficiency (*i.e.* DOC is likely masking the effect of increasing combustion efficiency on $\sum$SpVOC-IVOC emissions). These experiments also provide additional information on the effect of the DOC on the exhaust gas composition. The observation of increasing *n*-alkane abundance with increasing engine combustion efficiency, suggests that the increase in the abundance of the branched aliphatics at cold idle (exp. 14), 30% (exp. 12) and 40% engine load (exp. 13) respectively, is the result of the DOC fragmenting higher molecular weight branched aliphatics with increasing HC removal efficiency; indicating that the branched aliphatics are more easily fragmented in the DOC than *n*-alkanes, possibly the result of the fewer binding sites required in the DOC for adsorption.

### 3.5 Combustion and DOC removal efficiency

The measured $\sum$SpVOC-IVOC emission rates in each experiment (ordered from highest to lowest) are shown in Figure 11, along with the corresponding engine load, speed and temperature. The $\sum$SpVOC-IVOC emission rates varied significantly with different engine conditions, ranging from 70 to 9268 mg kg$^{-1}$. The aliphatics represented 56 to 97% of the $\sum$SpVOC-IVOC emission rates, with the single-ring aromatics accounting for the remainder. The highest $\sum$SpVOC-IVOC emissions were observed in a cold-start experiment (exp. 6) with no applied load and idling speed (1150 rpm). Conversely, the lowest $\sum$SpVOC-IVOC emissions were observed in the experiment with highest applied engine load and speed (exp. 11, 3000 rpm, 53% load). The $\sum$SpVOC-IVOC emission rates were observed to decrease with increasing engine load and temperature, and to a lesser degree, engine speed. This result is consistent with increased combustion efficiency and DOC HC removal efficiency with increasing engine temperature, similar to that observed in previous studies (*e.g.* (Cross et al., 2015; Chin et al., 2012)).

Urban driving conditions are characterised by low engine speed, load and exhaust gas temperatures (*c.f.* (Franco et al., 2014; EEA, 2016)). Conversely, motorway or highway driving typically result in higher engine temperatures, due to increased engine speed and load. The results from this study show at low engine loads and speeds, the emission rates of unregulated VOC-IVOCs per kilogram of fuel burnt, are considerably greater than emitted at higher engine speeds and loads (emission rates were 65 times greater from a cold-start than at maximum applied engine speed and load). Furthermore, it was found that the exhaust gas composition varied with combustion efficiency and DOC HC removal efficiency, both which are strongly dependent on working temperature (Korin et al., 1999; Roberts et al., 2014; Majewski and Khair, 2006; Russell and Epling, 2011). The effect of combustion efficiency on the exhaust gas composition was observed at engine temperatures below ~ 150°C (below the working temperature of the DOC) and at the maximum applied engine speed and load (700 °C), where combustion efficiency dominated over the effect of the DOC on the exhaust gas composition. At engine temperatures ranging from 150 - 450 °C, the combined effect of engine combustion and DOC HC removal efficiency on the exhaust gas composition was observed.

### 3.6 Discussion

Diesel exhaust emissions contain thousands of compounds ranging from ~ $C_5$ to $C_{22}$, with contributions of up to $C_{33}$ from lubricant oil (Alam et al., 2016; Gentner et al., 2017). Only a proportion of these emissions were speciated in this study. Of the measured compounds, branched aliphatics generally dominated the exhaust gas composition. An increasing contribution of branched aliphatics in the exhaust gas was observed with increasing engine temperature from ~ 150 - 450°C and is likely due to increasing DOC HC removal efficiency. However, below the working temperature of the DOC (< 150°C), the proportion of *n*-alkanes in the exhaust gas were observed to increase with increasing combustion efficiency and could be important in urban environments; straight-chain alkanes are more efficient at producing SOA than their branched counterparts (Presto et al., 2010; Tkacik et al., 2012; Lim and Ziemann, 2009). Previous studies have suggested liquid fuel based emission factors are consistent with unburnt fuel in diesel exhaust emissions. For example, Gentner et al. (2013) showed the majority of VOC and

IVOC diesel exhaust emissions were within 70% uncertainty of liquid fuel based emission factors. This work shows as combustion efficiency increases, the contribution of smaller, more volatile ($C_7$ to $C_{13}$) *n*-alkanes in the exhaust gas also increases, the result of increased fragmentation of higher molecular weight *n*-alkanes (> $C_{13}$, not measured) likely from the fuel and lubricate oil. This may suggest liquid diesel fuel based estimates of SOA yields may be inconsistent with diesel
exhaust SOA yields, particularly at high engine temperatures (*i.e.* high engine loads and speeds).

The comparison of emission rates is difficult between studies due to the vast number of differences (*e.g.* types of speciated compounds and volatility range, vehicular types, emission control devices *etc.*). Furthermore, few studies have investigated the effect of different engine conditions on the exhaust gas composition. The majority of studies have investigated diesel
exhaust emissions using chassis dynamometers, averaging emissions over entire driving cycles and often reporting emission rates as mass emitted per distance travelled; emissions and units which are not directly comparable with the emission rates shown in this work. The different types of instruments used to measure diesel exhaust emissions and the difficulties in the measurement of low volatility species, has in-part, resulted in considerable variation in the types of 'speciated' compounds between studies, further compounding the difficultly in the direct comparison of emission rates. For example, Zhao et al.
(2015) reported speciated and unspeciated IVOC emission rates from both medium-duty and heavy-duty diesel vehicles. In their study, speciated IVOCs included straight and branched chain alkanes, alkylcyclohexanes, unsubstituted and substituted polycyclic aromatic hydrocarbons and alkylbenzenes with a volatility range of $C^* $ $10^2$ to $10^6$ µg m$^{-3}$. Similarly, Cross et al. (2015) measured IVOC emission rates from a medium-duty diesel engine. The emission rates reported in their study were based on compounds with a volatility range of $C^* \sim 10^3$ to $10^7$ µg m$^{-3}$ and included cycloalkanes, bicycloalkanes,
tricycloalkanes, straight and branched chain aliphatics, and groupings of 'aromatics', 'oxidised' and 'remainder'. Gordon et al. (2015) measured the emission rates of non-methane organic gases from medium-duty and heavy-duty diesel engines. In their study, speciated compounds included single-ring aromatics, straight and branched chain aliphatics, cycloalkanes and non-aromatic carbonyls with a volatility range of $C^* \sim 10^6$ to $10^9$ µg m$^{-3}$. In this study, the emission rates from a light-duty diesel engine were reported, based on the emissions of straight and branched-chain aliphatics and single-ring aromatics with two and
three carbon substituents, with a carbon range of $C_6$ to $C_{13}$ and a volatility range of $C^* \sim 10^5$ to $10^8$ µg m$^{-3}$. The investigated chemical composition and volatility range can have a considerable impact on the reported emission rates (*e.g.* (Zhao et al., 2015)). The emission rates from comparable experiments in the studies discussed above can be observed in Table 3. However, a direct comparison of these emission rates with the results shown here, has not been performed due to the differences in vehicular type (medium-duty and heavy-duty *vs.* light-duty) and the volatility and chemical composition of the speciated
compounds. Further studies are required, providing emission rates of individually speciated compounds (where possible) to facilitate direct comparison.

Tailpipe sampling requires instruments capable of providing instantaneous measurements of the chemical composition. This rapid analysis time comes at the expense of detailed chemical speciation. The use of an atmospheric simulation chamber in

this study, allowed instruments requiring longer sampling times to be used, such as the GC×GC-FID. The chamber allowed detailed chemical speciation of the exhaust gas composition to be instantaneously measured under specific ambient temperatures and engine conditions, which would not have been possible with direct tailpipe sampling. Thus, the chamber sampling method is complimentary to tailpipe measurements, allowing a more thorough characterisation of the exhaust gas composition through the identification of individual hydrocarbon components. The emission control devices used in this study were Euro 4 compliant. Euro 4 emission control regulations were first implemented for all new vehicles from approximately January 2006, with Euro 5 emission control regulations starting post January 2011. Currently ~ 20% of the European Union diesel fleet are Euro 4 compliant (ACEA, 2017). The emission rates from only one diesel engine were investigated in this study. However, several compositional changes in the exhaust gas were comparable with previous studies, suggesting the effect of engine combustion and DOC HC removal efficiency on the exhaust gas composition, are relatively consistent between studies. In recent years, emission regulations have focused on reducing $NO_x$ emissions from diesel vehicles with the introduction of emission control technologies, such as exhaust gas recirculation (EGR), lean-burnt $NO_x$ traps and selective catalytic reduction (Yang et al., 2015). To our knowledge however, there are no further emission control technologies planned for the reduction of total hydrocarbon mass or unregulated VOCs. To reduce the effect of diesel exhaust emissions on local air quality, further technologies must be developed to reduce emission rates, specifically from cold engine conditions (*i.e.* poor combustion efficiencies) and below the working temperature of DOC. The experimental approach and results presented in this work will support further studies investigating the effect of different combustion engines, emission control devices and atmospheric conditions on the composition and evolution of exhaust gas emissions. To our knowledge, this is the first study using an atmospheric simulation chamber to separate the effects of the DOC and combustion efficiency on the exhaust gas composition.

**Acknowledgments**

The Manchester Aerosol Chamber previously received funding from the European Union's FP7 EUROCHAMP2 network and the Horizon 2020 research and innovation programme through the EUROCHAMP-2020 Infrastructure Activity under grant agreement no. 730997. The authors gratefully acknowledge the assistance of mechanical technicians Barry Gale and Lee Paul at the University of Manchester. This work was supported by NE/K012959/1 and NE/K014838/1.

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

**Table 1** – List of experiments performed, including the experimental dates, descriptions, chamber conditions and engine operating parameters.

| Experiment | Experiment Date | Experiment Description | Engine Conditions | | | | | | | Exhaust dilution ratio | Chamber Temp (°C) | Chamber RH (%) | Exhaust emission† | | |
|---|---|---|---|---|---|---|---|---|---|---|---|---|---|---|---|
| | | | RPM | Throttle (%) | Load (%) | Torque (Nm) | DOC | Engine Temp (°C) | Fuel Burnt (g) | | | | NO (g kg$^{-1}$) | NO$_2$ (g kg$^{-1}$) | Particle mass (mg kg$^{-1}$)‡ |
| 1 | 30.07.14 | Warm high load[a] | 2500 | 57 | 40 | 75 | Yes | 460 | 6.13 | 166 | 21.0 | 54.8 | 27.6 | 3.2 | 302 |
| 2 | 31.07.14 | Warm high load[a] | 2500 | 57 | 40 | 75 | Yes | 450 | 2.45 | 313 | 21.8 | 52.8 | 23.4 | 1.9 | 235 |
| 3 | 01.08.14 | Warm high load | 2500 | 57 | 40 | 75 | No | 450 | 2.45 | 325 | 31.6 | 61.6 | 21.7 | 2.5 | 198 |
| 4 | 05.08.14 | Warm with load[b] | 2000 | 40 | 30 | 50 | Yes | ** | 0.41 | 1158 | 32.9 | 52.0 | 26.1 | 0.5 | 220 |
| 5 | 08.08.14 | Warm with load[b] | 2000 | 40 | 30 | 50 | Yes | 300 | 8.27 | 60 | 32.6 | 58.7 | 20.7 | 3.1 | 268 |
| 6 | 06.08.14 | Cold Start[c] | 1150 | 0 | 0 | 1.5 | Yes | 85 | 0.59 | 389 | 32.7 | 58.8 | 7.2 | 15.0 | 1159 |
| 7 | 07.08.14 | Cold Start[c] | 1150 | 0 | 0 | 2 | Yes | 83 | 0.59 | 564*** | 27.4 | 52.8 | 7.9 | 14.6 | 915 |
| 8 | 06.08.14 (2) | Cold loaded* | 1500 | 30 | 20 | 32 | Yes | 169 | 1.19 | 312 | 27.5 | 51.8 | 17.3 | 9.8 | 121 |
| 9 | 06.08.14 (3) | Warm idle following load* | 1180 | 0 | 0 | 0.3 | Yes | 150 | 1.18 | 775 | 25.9 | 45.7 | 13.1 | 0.9 | 168 |
| 10 | 13.11.14 (1) | Warm with load[b] | 2000 | 40 | 30 | 50 | Yes | 300 | 0.41 | 840 | 22.4 | 53.2 | 31.5 | 2.0 | 351 |
| 11 | 13.11.14 (2) | High RPM, 53% load | 3000 | 75 | 53 | 112 | Yes | 700 | 3.28 | 353 | 23.3 | 53.4 | 21.7 | 1.9 | 2146 |
| 12 | 14.11.14 (1) | High RPM, 30% load | 3000 | 48 | 30 | 50 | Yes | 345 | 1.74 | 198 | 22.8 | 53.0 | 39.2 | 5.3 | 600 |
| 13 | 14.11.14 (2) | High RPM, 40% load | 3000 | 57 | 40 | 70 | Yes | 445 | 2.39 | 191 | 23.6 | 48.2 | 44.6 | 8.6 | 655 |
| 14 | 25.11.14 | Cold Start[c] | 1150 | 0 | 0 | 2 | Yes | ** | 0.59 | 564*** | 22.2 | 39.7 | 6.3 | 10.6 | 1159 |
| 15 | 01.10.15 | Warm with load[b] | 2000 | 40 | 28 | 50 | Yes | 292 | 1.57 | 331 | 31.5 | 56.4 | 12.8 | 0.5 | 298 |
| 16 | 29.09.15 | Warm with load[b] | 2000 | 40 | 28 | 50 | Yes | 293 | 1.57 | 337 | 28.0 | 67.1 | 12.2 | 0.5 | 241 |

Superscript letters a, b and c highlight replicate experiments using the same engine conditions. Fuel batch A used in experiments 1 to 9 and fuel batch B, used in experiments 10 to 16, see section 2.3 and 3.1.1 for further information. *= Sequence of engine conditions performed, see section 2.1 and SI Figure S1. **No engine temperature measurement (engine thermocouple non-responsive). ***Estimated exhaust dilution ratio based on pneumatic valve introduction time. †Expressed as emission rates (*i.e.* mass of emission per kg of fuel burnt). ‡Wall loss corrected.

**Table 2 –** Calculated diesel oxidative catalyst (DOC) hydrocarbon removal efficiency for the speciated VOC-IVOCs. Determined from measured emissions rates of the speciated VOC-IVOCs in two replicate experiments with (exp. 2, see Table 1) and without (exp. 3) a DOC.

| | Emission without catalytic converter (mg kg$^{-1}$) | Emission with catalytic converter (mg kg$^{-1}$) | Removal efficiency (%) |
|---|---|---|---|
| Individual Compounds | | | |
| Benzene | 19.50±1.75 | 1.88±0.17 | 90.4±9.0 |
| Toluene | 3.89±0.37 | 1.58±0.15 | 59.3±9.4 |
| Ethyl benzene | 1.56±0.36 | 0.05±0.01 | 97.1±22.8 |
| m/p-xylene | 2.98±0.62 | 0.13±0.03 | 95.7±20.9 |
| o-xylene | 2.17±0.55 | 0.22±0.06 | 89.7±25.3 |
| Styrene | 2.74±0.69 | 0.01±0.004 | 99.5±25.3 |
| 1,3,5-TMB | 2.26±0.36 | 0 | 100* |
| 1,2,4-TMB | 2.45±0.21 | 0 | 100* |
| 1,2,3-TMB | 1.92±0.20 | 0 | 100* |
| Heptane | 2.37±0.14 | 0.53±0.03 | 77.4±5.7 |
| Octane | 4.96±0.57 | 0 | 100* |
| Nonane | 11.72±1.08 | 2.44±0.22 | 79.2±9.2 |
| Decane | 33.83±3.11 | 7.30±0.67 | 78.4±9.2 |
| Undecane | 49.76±4.57 | 32.34±2.97 | 35.0±9.2 |
| Dodecane | 137.65±12.64 | 156.60±14.38 | 0** |
| Groupings | | | |
| Branched aliphatics | | | |
| C$_7$ | 4.41±1.00 | 1.25±0.28 | 71.6±22.6 |
| C$_8$ | 18.77±4.76 | 3.68±0.93 | 80.4±25.4 |
| C$_9$ | 46.78±10.70 | 6.65±1.52 | 85.8±22.9 |
| C$_{10}$ | 76.81±18.78 | 17.33±4.24 | 77.4±24.4 |
| C$_{11}$ | 71.95±16.43 | 31.97±7.30 | 55.6±22.8 |
| C$_{12}$ | 86.36±20.80 | 74.41±17.92 | 13.8±24.1 |
| Aromatic substitutions | | | |
| C$_3$ | 14.18±3.13 | 5.20±1.15 | 63.3±22.1 |
| Total groupings | | | |
| Aliphatics | 545.37±37.22 | 334.50±24.74 | 38.66±11.7 |
| Aromatics | 53.65±3.81 | 9.07±1.17 | 83.09±2.6 |
| Total speciated | 599.02±37.41 | 343.58±24.77 | 42.64±9.7 |

* Compound not observed (< instrument LOD). **No observed decrease in concentration. TMB = trimethyl benzene. Errors
5   represent the calculated experimental uncertainty, see SI for further information.

**Table 3** – Literature emission rates (expressed as mass emitted per kg of fuel burnt) from diesel exhaust emissions using different vehicle types and engine conditions.

| Reference | Compounds Studied | Vehicle Type | Engine Size (L) | Emission Control Devices | Fuel Type | Driving Cycle | Load (%) | Speed (rpm) | Emission Rate (mg kg$^{-1}$) |
|---|---|---|---|---|---|---|---|---|---|
| Cross et al. (2015) | IVOCs | MDDV | 5.9 | None | ULSD | - | 0 | idle | 220 |
| Gordon et al. (2014) | NMOG | HDDV | 10.8 | None | USLD | UC | - | - | 6100 |
| | | MDDV | 6.6 | DOC | ULSD | UDDS | - | - | 1000 |
| | | MDDV | 5.9 | None | USLD | UC | - | - | 700 |
| Zhao et al. (2015) | IVOC | MDDV* | 12.8, 14.9 | DPF | USLD | Creep + idle | - | - | 600 |
| | | MDDV* | 12.8, 14.9 | DPF | USLD | UDDS, UC, Hi-Cruse | - | - | 20 |
| | | MDDV* | 6.6, 5.9, 10.8 | None | USLD | Creep + idle | - | - | 4000 |
| | | MDDV* | 6.6, 5.9, 10.8 | None | USLD | UDDS, UC, Hi-Cruse | - | - | 700 |
| | | Off-road LDDV | 2.2 | None | ULSD | 4-mode EPA TRU | - | - | 700 |
| Gordon et al. (2013) | NMOG | Off-road LDDV | 2.2 | None | ULSD | 4-mode EPA TRU | - | - | 2000 |

* = average emission rate of multiple experiments and vehicles with similar engine and driving cycle conditions, see Zhao et al. (2015) for further information. UC = unified driving cycle. UDDS = Urban dynamometer driving schedule. EPA TRU = Environment protection agency transportation refrigeration unit, see Gordon et al. (2013) for further information.

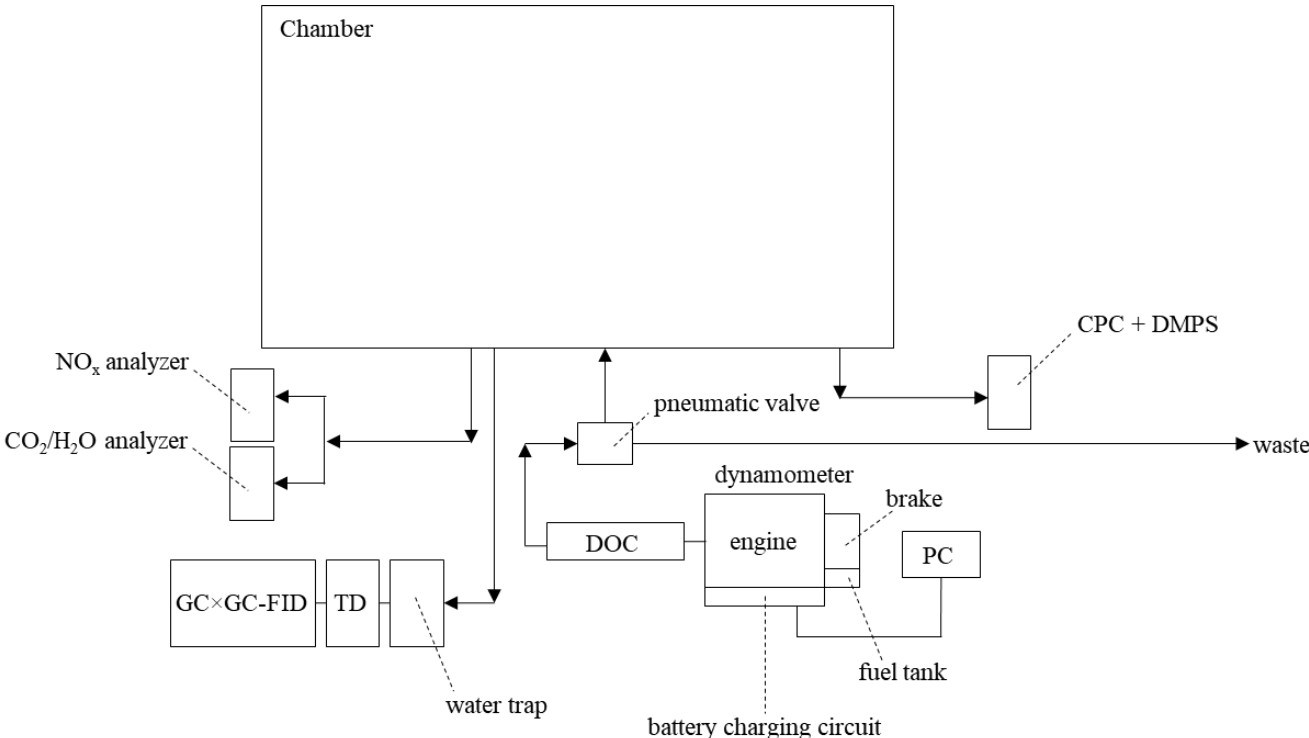

**Figure 1** – Schematic of dynamometer and sampling system. Arrows display air flow direction. CPC = condensation particle counter. DMPS = differential mobility particle sizer. DOC = diesel oxidative catalyst. TD = thermal desorption unit. GC×GC-FID = comprehensive two-dimensional gas chromatography flame ionisation detector. See Section 2.3 for further information.

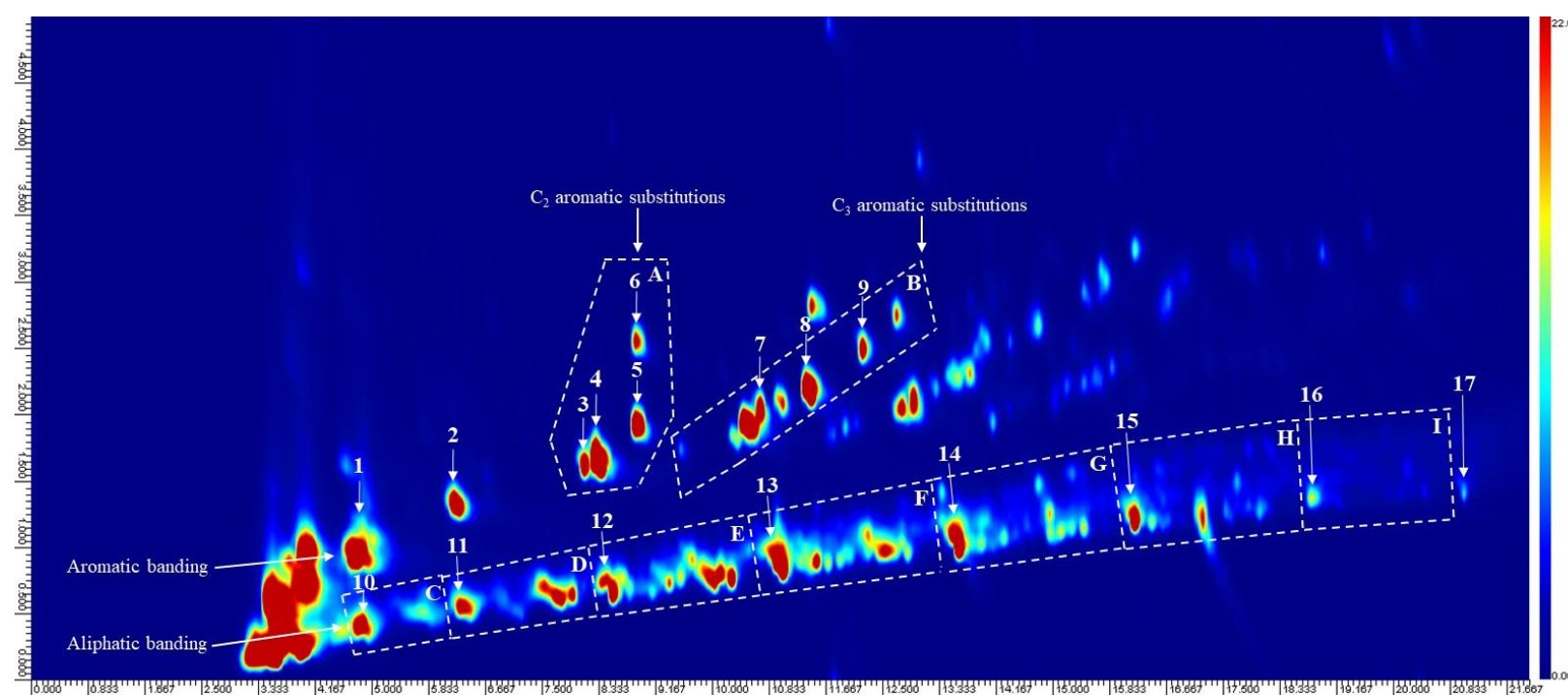

**Figure 2 –** An annotated chromatogram displaying the speciated VOC-IVOCs. Chromatogram axis, x = first dimension separation (boiling point, increasing from left-to-right), y = second dimension separation (polarity, increasing from bottom-to-top). Colour scale represents peak intensity, increasing from blue to red. Letters refer to compound groupings; A = single-ring aromatics with two carbon substitutions, B = single-ring aromatics with three carbon substitutions, C to I = $C_7$ to $C_{13}$ aliphatics grouped by carbon number (*i.e.* C = $C_7$ aliphatics, D = $C_8$ aliphatics *etc.*). Numbers refer to individual compounds; 1 = benzene, 2 = toluene, 3 = ethyl benzene, 4 = meta/para-xylene (co-elution), 5 = ortho-xylene, 6 = styrene, 7 = 1,3,5-trimethyl benzene, 8 = 1,2,4-trimethyl benzene, 9 = 1,2,3-trimethyl benzene, 10 = heptane, 11= octane, 12 = nonane, 13 = decane, 14 = undecane, 15 = dodecane, 16 = tridecane, 17 = tetradecane (not quantified). Aromatic and aliphatic bandings often observed with this technique are shown (*c.f.* Hamilton and Lewis (2003) and Dunmore et al. (2015)). The start and end of each aliphatic grouping is marked by the lower and higher carbon number *n*-alkane (*i.e.* nonane marks the start of the $C_9$ aliphatic grouping, decane marks the end of this group).

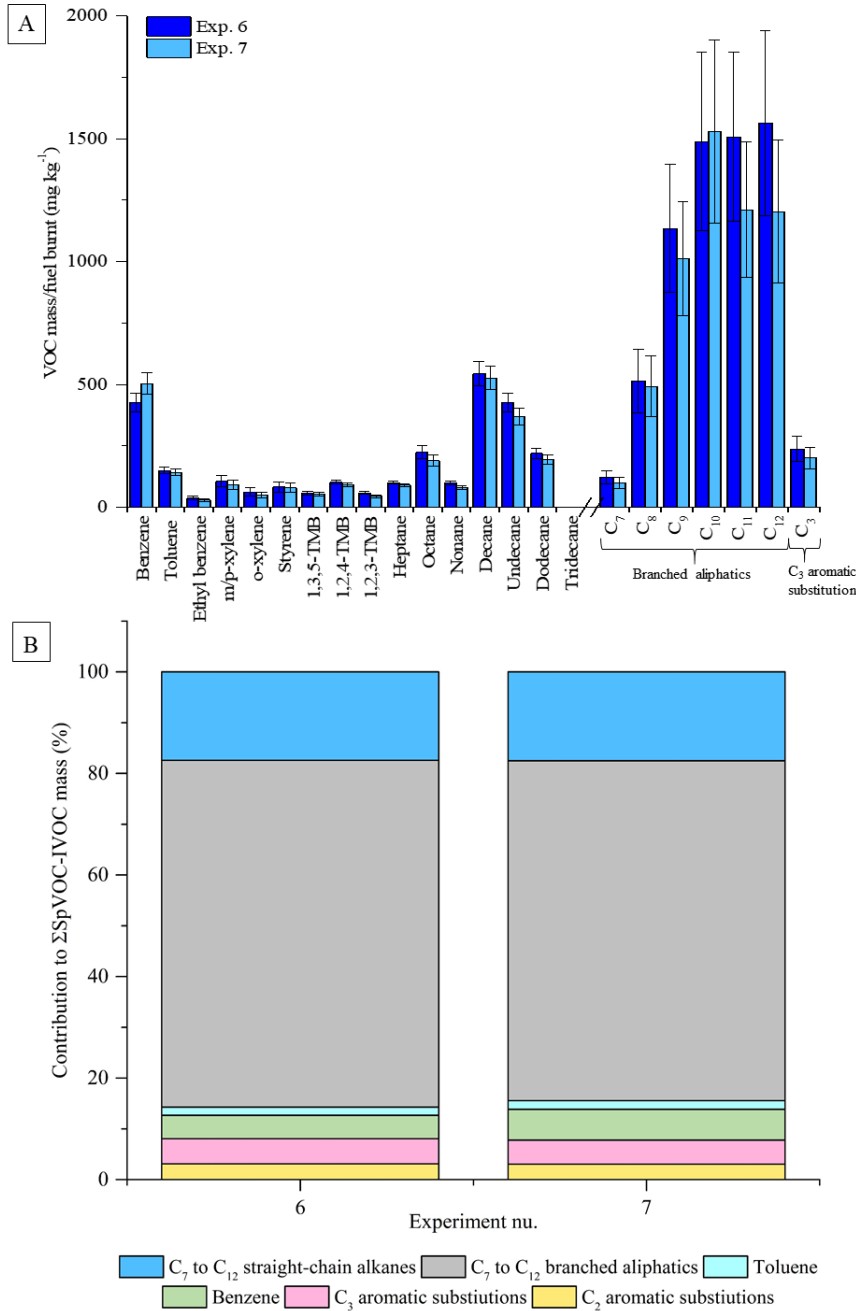

**Figure 3** – Comparison of measured VOC-IVOC emission rates in replicate cold-start experiments (exp. 6 and 7) (A). Comparison of the percentage contribution of the individual and grouped compounds to the $\sum$SpVOC-IVOC emission rates in exp. 6 and 7 (B). The emission rates of tridecane and the $C_{13}$ branched aliphatic grouping has not been included in (B) to allow direct comparison between other experiments where these species were not measured. Error bars represent the calculated uncertainty in the measured emission rates, see the SI section 1.1 for further information.

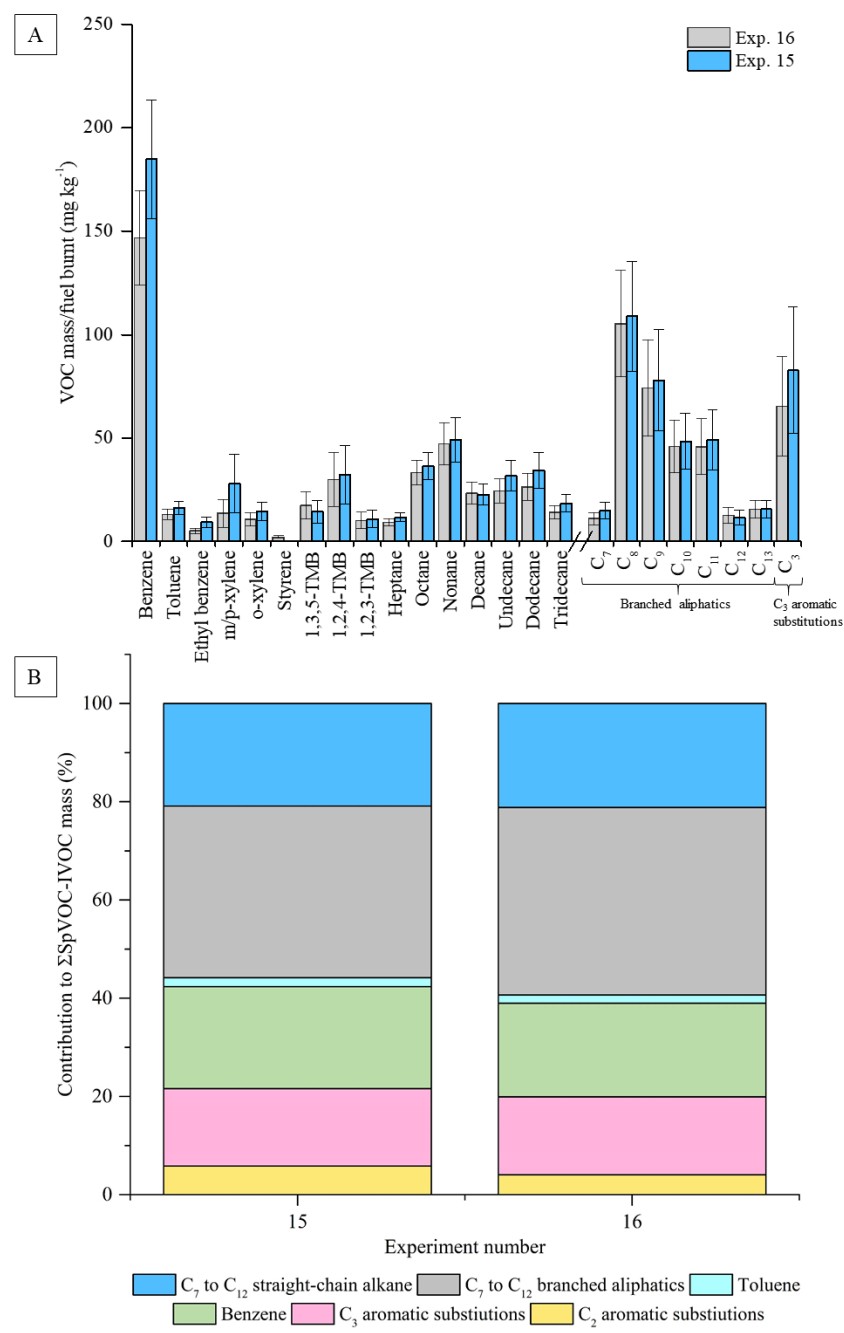

**Figure 4** – Comparison of measured VOC-IVOC emission rates in replicate WWL experiments (exp. 15 and 16, 2000 rpm, 30% load) (A). Comparison of the percentage contribution of the individual and grouped compounds to the ∑SpVOC-IVOC emission rates in exp. 15 and 16 (B). The emission rates of tridecane and the $C_{13}$ branched aliphatic grouping has not been included in (B) to allow direct comparison between other experiments where these species were not measured. Error bars represent the calculated uncertainty in the measured emission rates, see the SI section 1.1 for further information.

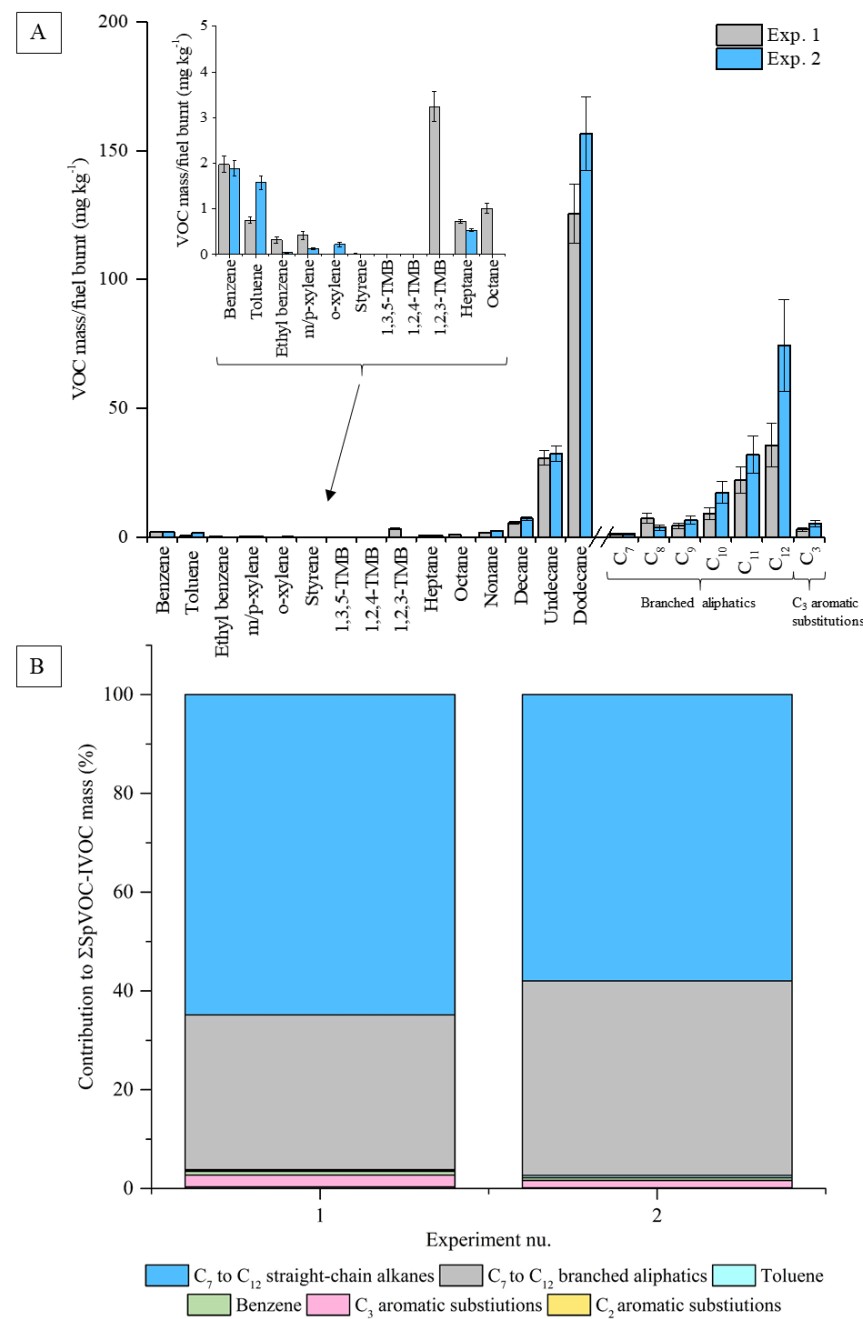

**Figure 5** – Comparison of measured VOC-IVOC emission rates in replicate warm high load experiments 1 and 2 (exp. 1 and 2, 2500 rpm, 40% load) (A). Comparison of the percentage contribution of the individual and grouped compounds to the $\sum$SpVOC-IVOC emission rates in exp. 1 and 2 (B). The emission rates of tridecane and the $C_{13}$ branched aliphatic grouping has not been included in (B) to allow direct comparison between other experiments where these species were not measured. Error bars represent the calculated uncertainty in the measured emission rates, see the SI section 1.1 for further information.

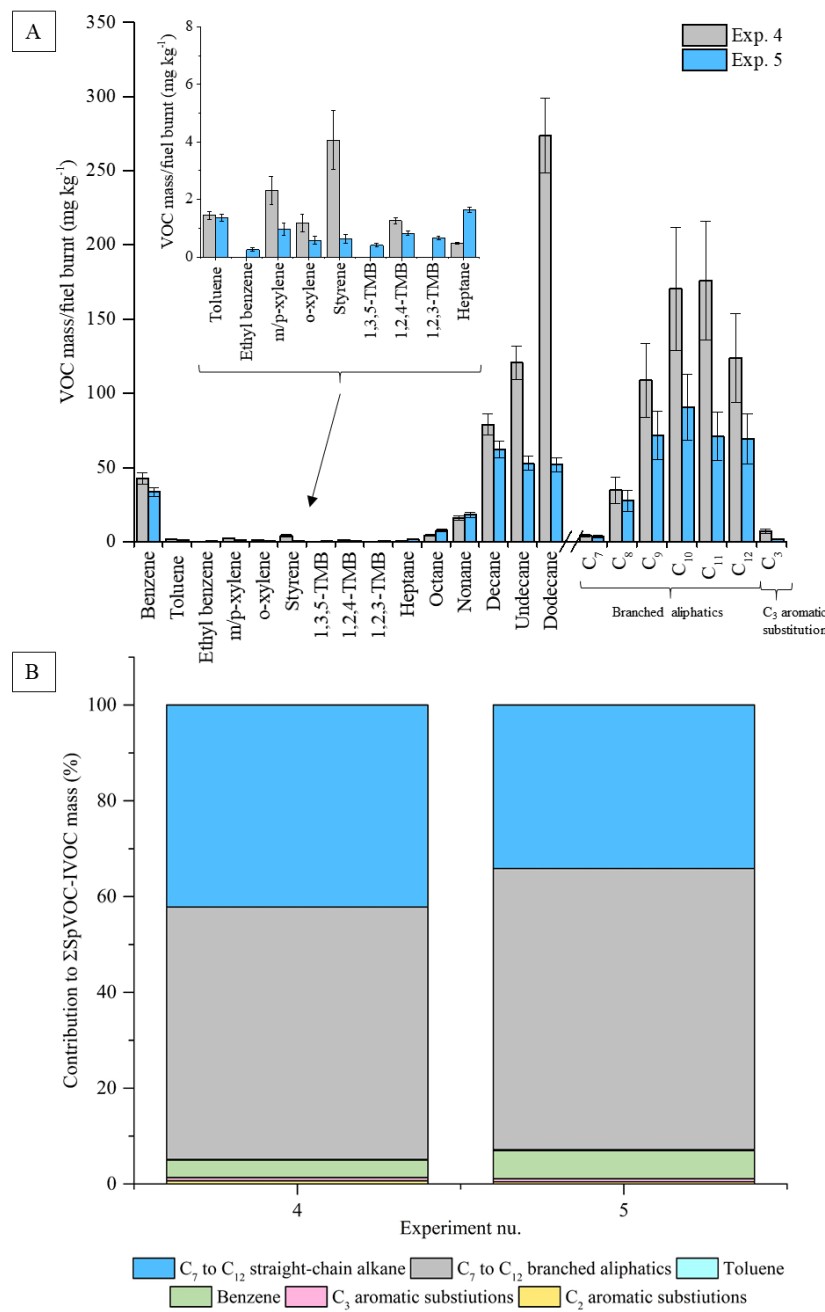

**Figure 6** – Comparison of measured VOC-IVOC emission rates in replicate warm with load experiments 4 and 5 (2000 rpm, 30% load) (A). Comparison of the percentage contribution of the individual and grouped compounds to the $\sum$SpVOC-IVOC emission rates in exp. 4 and 5 (B). The emission rates of tridecane and the $C_{13}$ branched aliphatic grouping has not been included in (B) to allow direct comparison between other experiments where these species were not measured. Error bars represent the calculated uncertainty in the measured emission rates, see the SI section 1.1 for further information.

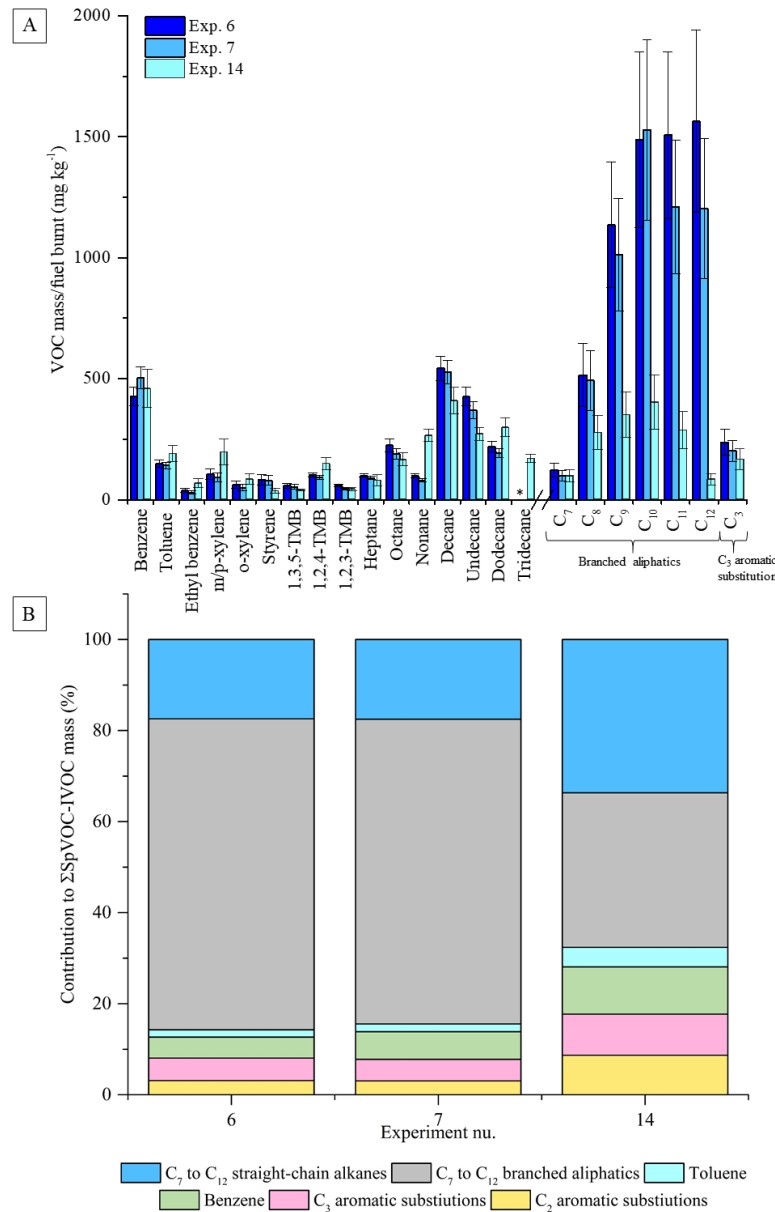

**Figure 7** – Comparison of measured VOC-IVOC emissions rates in replicate cold-start experiments 6 and 7 (fuel batch A) with cold-start experiment 14 (fuel batch B) (A). Comparison of the percentage contribution of the individual and grouped compounds to the ∑SpVOC-IVOC emission rates in experiments 6, 7, 14 (B). The emission rates of tridecane and the $C_{13}$ branched aliphatic grouping has not been included in (B) to allow direct comparison between other experiments where these species were not measured. Error bars represent the calculated uncertainty in the measured emission rates, see the SI section 1.1 for further information.

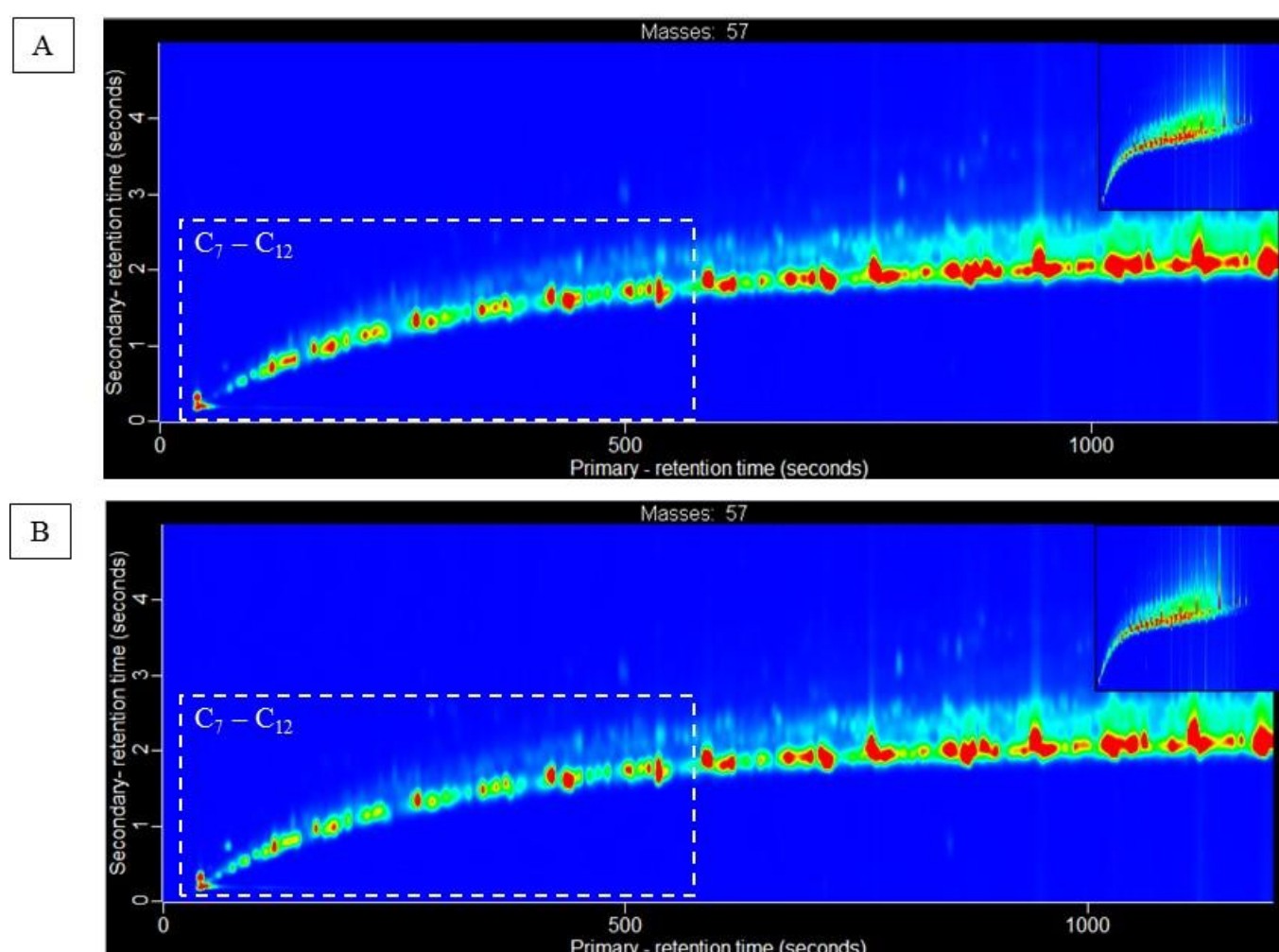

**Figure 8** – Extracted ion chromatogram of *m/z* 57 (dominate aliphatic fragment ion) for the liquid diesel fuel samples analysed using GC×GC-TOFMS. A = Fuel batch A (see section 3.1.1 for further information). B = Fuel batch B. Chromatogram axis, x = primary column, first dimension separation (boiling point, increasing from left-to-right), y = secondary column, second dimension separation (polarity, increasing from bottom-to-top). Colour scale represents peak intensity, increasing from blue to red. Chromatograms have been normalised to allow direct comparison of peak intensity between chromatograms. Dashed box highlights an approximate carbon number range of $C_7$ to $C_{12}$, determined from the NIST library identification of individual compounds.

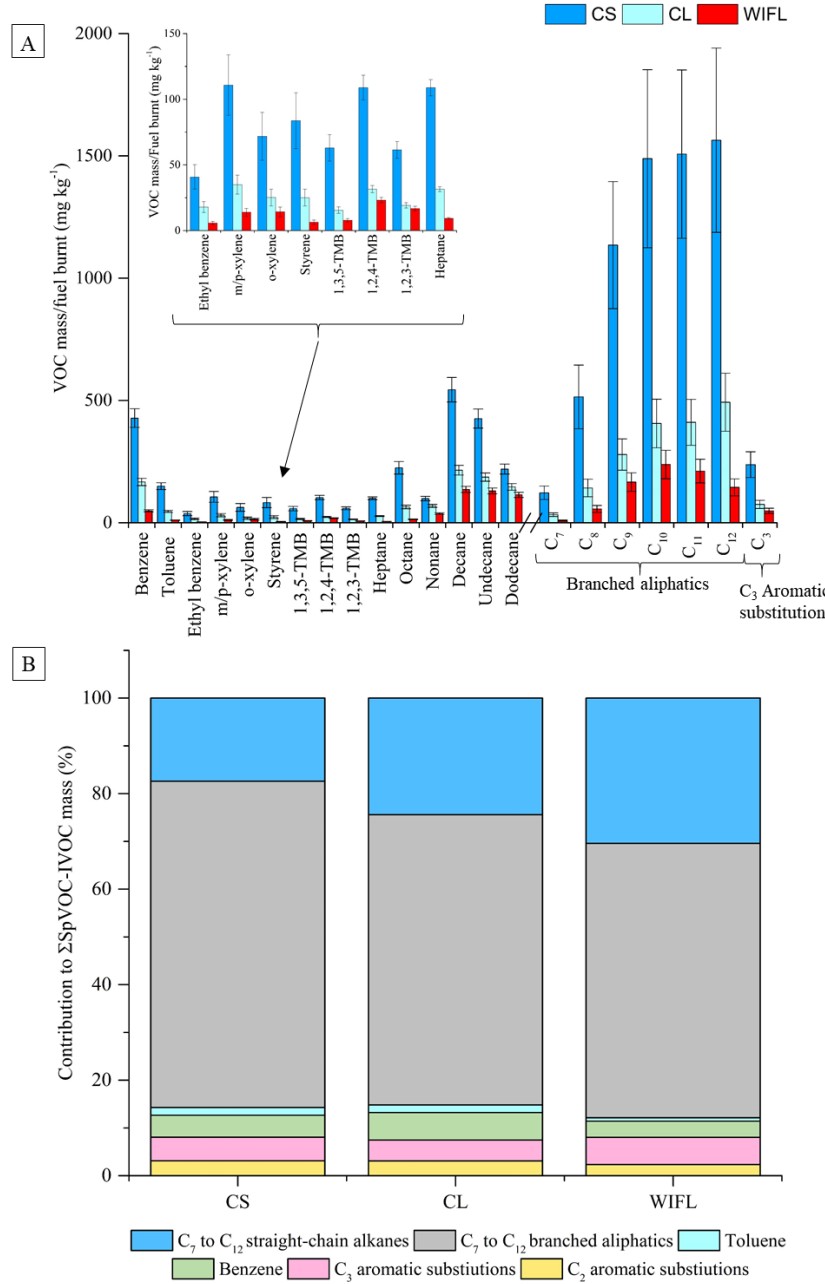

**Figure 9** – Effect of different driving scenarios on measured VOC-IVOC emission rates (A) and the contribution of the individual and grouped compounds to the $\sum$SpVOC-IVOC emission rate (B). CS = cold-start (exp. 6). CL = cold loaded (exp. 8), WIFL = warm idle following load (exp. 9, see text for further information). The emission rates of tridecane and the $C_{13}$ branched aliphatic grouping have not been included in (B) to allow direct comparison between other experiments where these species were not measured. Error bars represent the calculated uncertainty in the measured emission rates, see the SI section 1.1 for further information.

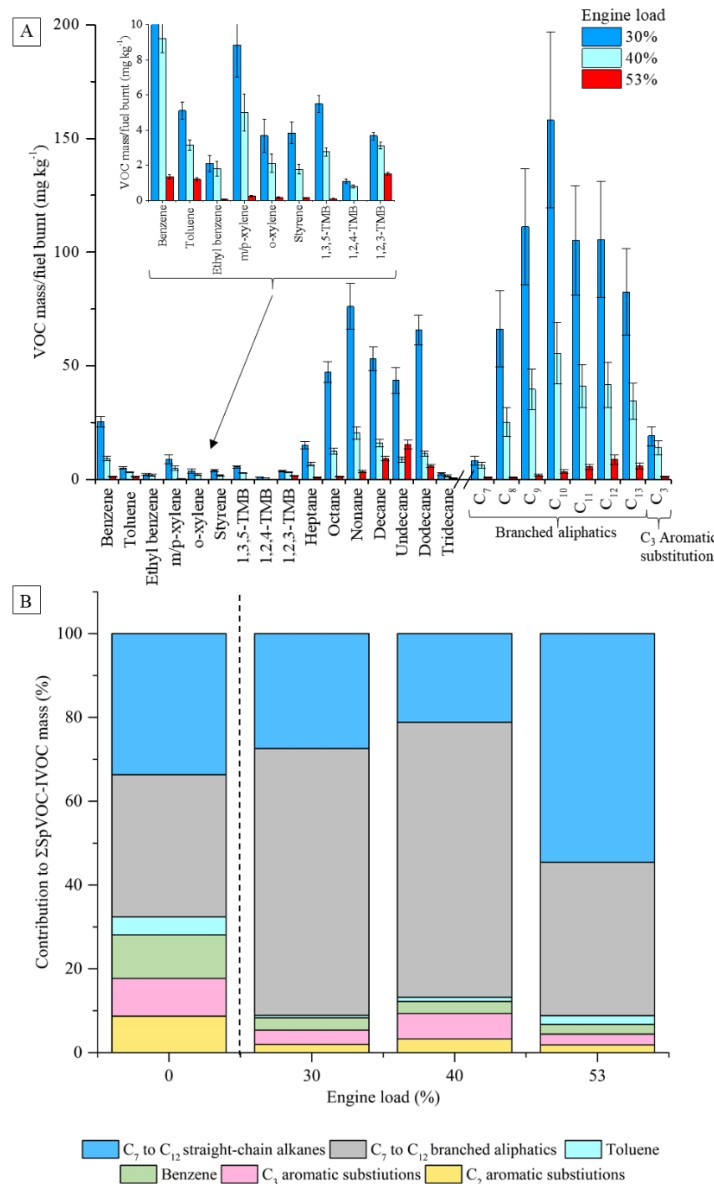

**Figure 10** – Effect of different engine loads on measured VOC-IVOC emission rates (A) and the percentage contribution of the individual and grouped compounds to the $\sum$SpVOC-IVOC emission rate at 0 (exp. 14), 30 (exp. 12), 40 (exp. 13) and 53% (exp. 11) engine load (B). The emission rates of tridecane and the $C_{13}$ branched aliphatic grouping have not been included in (B) to allow direct comparison between other experiments where these species were not measured. For comparison, the percentage contribution of the individual and grouped compounds to the $\sum$SpVOC emission rate in a cold idle experiment (exp. 14) has been included on the left of (B), see text for further details. Error bars represent the calculated uncertainty in the measured emission rates, see the SI section 1.1 for further information.

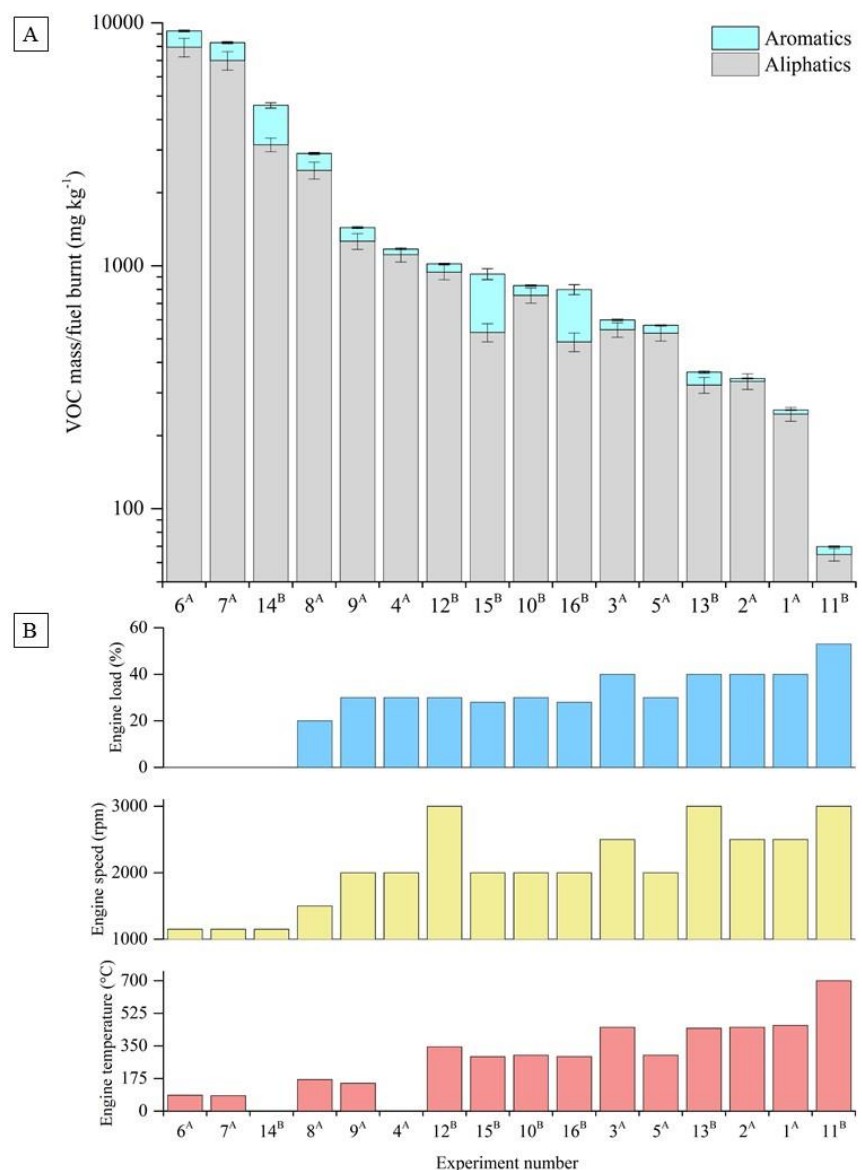

**Figure 11 –**Total speciated VOC-IVOC emission rate measured in each experiment (refer to Table 1) divided into aliphatic and aromatic emissions rates (A). Experiments ordered from left-to-right by decreasing VOC-IVOC emission rates. Engine temperature, speed and load in each corresponding experiment is shown in (B). A = Fuel batch A used (see sections 2.3 and 3.1.1). B = Fuel batch B used. No DOC in exp. 3. Sequence of engine conditions performed in exp. 9 and 8 (see section 2.1 and SI Figure S1). No engine temperature measurement for exp. 4 and 14 (engine thermocouple unresponsive). Error bars represent the calculated uncertainty in the measured emission rates, see the SI section 1.1 for further information.