# Peer review of "Technical Note: Use of an Atmospheric Simulation Chamber to Investigate the Effect of Different Engine Conditions on Unregulated VOC-IVOC Diesel Exhaust Emissions"

_Atmospheric Chemistry and Physics, 2017_

## Referee Comment (RC1) · Anonymous Referee #2 · 27 Sep 2017

General Comments

This manuscript presents diesel exhaust measurements of from a light-duty diesel engine using an engine dynamometer. The study investigated the effects of engine load and use of diesel oxidative catalyst (DOC) on emissions of volatile and intermediate volatility organic compounds (VOCs/IVOCs) in diesel exhaust. The exhaust emissions were diluted in an atmospheric chamber and online sampling was conducted to measure speciated VOCs and grouped VOCs using a TD-GCxGC-FID system. The study results showed that VOC/IVOC emission rates and VOC profiles were greatly impacted by engine load and DOC efficiency. Intermediate volatility organic compounds (IVOCs) are quite difficult to sample and quantify accurately, and speciated IVOC emissions measurements are rare, particularly online measurements. Therefore, the publication

of results from this work will be of significant benefit to the emissions research community to improve our understanding of IVOC emissions from a diesel engine and how they are impacted by engine conditions. However, there are several important issues raised below that must be addressed before this work is accepted for publication in ACP.

One major issue is that the rationale behind selection of the engine, fuel, experimental conditions was not articulated in the manuscript. Why was the particular engine with retrofitted DOC selected, and why was the goal to mimic Euro 4 emissions standard? I would imagine this engine/technology must be outdated compared to current and near future European light-duty vehicle fleet adhering to Euro 5 and 6 standards. Therefore, are these emission rate results at all relevant to the current European vehicle fleet?

Why were the specific experimental conditions chosen: specific engine loads, dilution ratios, speeds, multiple fuel batches? Based on the experiments listed on Table 1, the research strategy behind these measurements is very difficult to tease out. For example, in Section 3.1 the effects of engine load were discussed. However, because the DOC was also included, the results reflected the coupled effects of engine load and DOC that were interdependent. This is not ideal for a mechanistic study such as this work. Was it intended to study the effects of both simultaneously or was this an unintended consequence of the experimental design? Either way, no explanation was given. The rationale behind the experimental design and study conditions needs to be clearly described in order for the reader to interpret the results.

The second major issue is that a large portion of the discussion in the supplementary information is extremely important and should be provided in the main text. I believe some discussion points brought up in the SI are actually more scientifically relevant and impactful than some of the extended discussion about engine load and DOC effects in the main text that have been previously studied by others.

Experimental reproducibility is vital experimental information; this information (e.g.

measurement replicates, test replicates) needs to be clearly detailed in the main text in order for the reader to assess the data quality and statistical power of the results. In Table 1, which experiments were meant to be replicate tests, and are the numbers in parentheses the measurement replicates or something else? Also, the origins of the error bars and emission rate standard deviation values need to be explained in the main text. If they are from the calibration propagation of error uncertainty calculations in SI Section 3.1, then how were measurement and test replicates included in the errors?

The discussion in the SI indicated that dilution ratio had a substantial impact on IVOC emissions between two similar experiments. Given the wide range of dilution ratios used in this study, this issue warrants further discussion in the main text, particularly how this issue may have influenced the results. Furthermore, a discussion of the observed differences in emission rates for the two different fuel batches and linking them to fuel composition by 2DGC-TOF fuel analyses was very interesting and should be included in the main text.

The third major issue is that in Section 3.4 the comparison between these engine dynamometer emission rates with California tunnel measurements is misleading and not scientifically relevant. While I understand the desire to link these results to real world measurements, I don't see the scientific merit or rationale for making this particular comparison. This study used an engine, aftertreatment technology, and fuel that are not at all relevant for these CA on-road measurements. What I believe would be of much greater scientific benefit to the mobile source emissions research community is a thorough literature comparison of speciated IVOC measurements that are so rarely measured for at least light-duty diesel engines and vehicles from engine and chassis dynamometer studies. Are these measurements consistent with other light-duty (and perhaps also medium/heavy duty) diesel vehicles and how to they compare with Euro5 and Euro6 compliant engines/vehicles/fuels? If the IVOC data is lacking for modern vehicles with newer emission standard, how are the emissions expected to change from Euro 4 to Euro 5 and 6. Finally, how will the engine operation and other effects on

VOC/IVOC emissions impact SOA formation from vehicle exhaust?

Specific Comments

Section 1. Introduction: there are some repetitive statements (Page 2, Lines 17-20; Page 3, Lines 34-36; Page 4, Lines 1-2, 5-7). The introduction could be made more concise by removing the repetitive text.

Page 4, Lines 10-12. The introduction referenced research that has previously studied the effects of DOC and engine operating parameters on vehicle exhaust composition, e.g. Chin et al. 2012. Other studies have also investigated these effects on diesel vehicles/engines (e.g. Ballesteros et al. (2014), Zhu et al. (2013)). Can this statement be further clarified so that it is more accurate?

Page 4, Line 28. Please provide hygrometer vendor information.

Section 2.2. Section title is misleading. This section not only describes the engine but also exhaust sampling system. Please provide more detailed information on the engine specifications and about DOC usage status (mileage or rapid thermal aging hours) as this can significantly impact the emissions. Provide rationale behind the selection of the specific engine, aftertreatment and why Euro 4 emission control was approximated. Please provide a schematic of the dynamometer and sampling system (this may be included in the SI). Please describe all engine operating conditions including driving scenarios and include rationale behind the experimental design.

Page 5, Line 12-13. Was the MAC filled before exhaust was introduced? How were the dilution ratios decided upon and why did they vary from experiment to experiment? How long was the diluted exhaust allowed to equilibrate before sampling took place and were there any apparent losses in IVOCs over time?

Page 5, Line 18. Please explain why two batches of diesel fuel were used. Please include any fuel composition or fuel property analyses that were conducted. What was the sulfur content of the two batches? Do both batches of the diesel fuel meet Euro 4

or Euro 5 specifications? Please discuss differences in fuel composition here. What is meant by "standard"?

Page 5, Line 25. What temp was the line heated to?

Page 6, Line 17. How often was the calibration checked by a standard? Were MDLs determined? If so, explain how and provide MDL values for all compounds. This can be provided in the SI.

Page 7, Line 1-4. The engine operating conditions should be described in detail in the experimental as well as justification for why these specific conditions were chosen.

Page 7, Line 9. Not enough evidence has been provided to identify the grouped VOCs "branched aliphatics", particularly because these were not GC/MS measurements. These would be better labeled as unspeciated or unresolved organics. The grouped VOCs should also not be called speciated VOCs because they are not individually speciated. Therefore, sum of SpVOC is a misleading term and should also be changed to minimize confusion. Also, please discuss why this narrow range of VOCs was studied? Was it intentional to focus on IVOCs or was this due to sampling/instrumental limitations?

Page 7, Line 11. How often were the retention time windows confirmed? Were any internal standards used?

Page 7, Line 23-24. Both reproducibility and dilution effects are important enough to merit detailed discussion in the main text. Please see general comments above.

Page 7, Line 29-30. While similar sets of conditions have been compared in the discussion, emission factors are all presented for every experiment, allowing the reader to freely compare between experiments without fully understanding the impacts of differing dilution ratios unless they read the SI in detail. This is why a full discussion of the effects of dilution ratios is needed in the main text.

Section 3.1 Why were these loads chosen? Please state clearly whether these experiments (and for each subsection in the Results section and in all figure captions) are from tests with or without DOC? This section title is also misleading because the combined effects of engine load and DOC were studied simultaneously.

Page 8, Line 7. Is this statement suggesting that engine load is the most important factor that controls VOC emission rates, whereas other factors, e.g. aftertreatment and fuel, are less important? If not, please clarify. Also, please state clearly that the assumption was made that engine load is linearly related to engine combustion efficiency or provide ancillary measurements to confirm combustion efficiencies for each experiment.

Page 8, Line 18, 20. Should the reference be Chin et al. (2012)?

Page 9, Lines 16-22. This text contradicts what is shown in Figure 2a, 2b and previous discussion on Page 8.

Page 10, Lines 16-17. Did Alam et al. use the same type of DOC or was it different from what was used in this study? It may also be interesting to compare the results of Liu et al. (2010) with this work.

Page 10, Lines 26-30. Please show how the conditions of these three scenarios change in a diagram in the experimental section or SI.

Page 11, Line 17-18. Relative abundances discussed here?

Page 11, Line 18-22. It is not clear why the effect of the DOC are brought up here. It does not logically fit with the rest of the discussion in this paragraph.

Section 3.4. Please see general comments on this section. It would be extremely useful to conduct a literature comparison to assess whether the emission factors from this study are consistent with similar studies. The comparison with Gentner et al. (2013) is misleading and should not be included.

Page 12. Line 32-33. Aldehydes are not at all mentioned here and represent arguably

the most important air toxic/hazardous VOCs emitted from modern diesel vehicles.

Page 13. Lines 3-7. Please discuss differences between Euro 5 and Euro 6 emission standards in relation to the approximated Euro 4 emission standards simulated here and how modern vehicles and their aftertreatment technologies are specifically changing. How are these changes expected to impact VOC emissions? Emission control technologies targeting other pollutants may still have significant effects on VOC emissions. This is where the literature comparison of emission factors from this study with emission studies with Euro 5 and Euro 6 vehicles would provide insight (see general comments).

Page 13. Lines 7-8. Please see general comments about this comparison.

Page 13. Lines 11-13. Please see earlier comment on the veracity of this statement.

Table 1. Please clearly indicate which tests were replicate tests and which tests had replicate measurements.

Figure 1. What are the compounds below B and above G? What is the peak above #8? Why does benzene elute after toluene – are those labels correct? Have any attempts been made to quantify the lighter compounds and to assess total unspeciated organics?

Figure 2. What do error bars represent? Are these tests with or without DOC? Toluene and benzene colors are difficult to distinguish.

Figure 4. Evidence was not provided to speciate into aliphatic and aromatic components.

Figure 5. See general comments on this comparison.

Supplementary Information

Page 4, Lines 13-14. Why was toluene used and not an n-alkane?

References:

Ballesteros, R., Guillen-Flores, J. Martinez, J. D.: Carbonyl emission and toxicity profile of diesel blends with an animal-fat biodiesel and a tire pyrolysis liquid fuel, Chemosphere, 96, 155–166, 2014.

Liu, Z. G., Berg, D. R., Vasys, V. N., Dettmann, M. E., Zielinska, B., Schauer, J. J.: Analysis of C1, C2, and C10 through C33 particle-phase and semi-volatile organic compound emissions from heavy-duty diesel engines, Atmospheric Environment, 44 1108-1115, 2010.

Zhu, L., Cheung, C.S., Zhang, W.G., Fang, J.H., Huang, Z.: Effects of ethanol– biodiesel blends and diesel oxidation catalyst (DOC) on particulate and unregulated emissions, Fuel, 113, 690–696, 2013.

---

## Referee Comment (RC2) · Anonymous Referee #1 · 6 Oct 2017

Some comments on "The effect of varying engine conditions on unregulated VOC diesel exhaust emissions"

In Europe almost half of all new passenger cars are diesel vehicles. The large number of diesel vehicles means that their emissions are an important source of air pollution in urban environments. This study focuses on speciated VOCs, including 16 individual and 8 groups of compounds and effects of a home-retrofitted DOC on the mass emissions and chemical composition of these VOCs from an older diesel engine. VOCs contribute to less than half of the organics in diesel exhaust with these specific compounds contributing an even smaller fraction (e.g. classic paper Schauer et al. EST 1999 or more recent papers by Gentner PNAS 2013 or Zhao et al. EST 2015). The majority of organic emissions from diesel vehicles are IVOCs. This paper does not

provide this important context.

The major weakness of the paper is that there is no substantive connection between the emissions data and air quality. Instead the paper focus on engine operations and emissions (an interesting topic but it seems outside the scope of ACP). Such a connection seems important for publishing in an atmospheric science journal. Therefore this study seems poorly suited to Atmospheric Chemistry and Physics.

The paper makes numerous claims about the novelty and importance of the work. Many papers have examined VOC speciation of diesel exhaust and robust VOC speciation profile exist for diesel exhaust (the major problem with these profiles is the lack of IVOC data). The paper provides a very limited review of this literature and some readers may be confused on state of knowledge of diesel VOC emissions after reading the intro.

The paper states multiple times that few studies have reported speciated emissions as a function of engine conditions. It is true that less is known about speciation as function of engine load and DOC then cycle based emissions (but much more work has been published than cited by this paper; a super quick search revealed multiple papers including Combust Flame, 118, 179, 1999; Atmos Env 42, 769, 2008, etc.). This paper, similar to the previously published work, clearly shows variations with engine loads and control technologies. The general trends (e.g. higher emissions at lower loads and changes in composition with loads) are consistent with the published literature. Similarly the results for the effectiveness of the DOC are similar to other studies (though at the low end of effectiveness presumably due to the retrofit nature of this application). More data are always good but the results are not especially novel from an emissions perspective. What are the implications of this new data from an atmospheric perspective (the focus of ACP)?

Air quality impacts depend on the integrated emissions (from many engines operated over a wide range of load conditions). The purpose of test cycles is to measure representative emissions that are relevant to atmosphere (emissions models like MOVES are moving to a more dynamic representation). Is there some problem with existing diesel VOC emissions profiles used by models and inventories that this paper is addressing? That was not clear (nothing jumps out to me), but I don't see an atmospheric question this paper is addressing. It seems more like engine / control technologies related questions. Maybe the paper belong in a different journal?

The study has used an advanced instrument (2D-GC-MS). However, the much of the analysis focuses on emissions of a commonly characterized subset of VOCs (n-alkanes and aromatics) that contribute a minor fraction of emissions. In addition, they are not the dominant source of SOA (e.g. dominated by IVOCs) or toxicity (carbonyls) from diesels. Not leveraging more the advanced 2D GC data seemed like a potentially missed opportunity.

The engine is operated over extremely simple test cycles (e.g. constant speed and load or idle followed by constant load for a number of minutes). This is very unrepresentative of essentially all actual in-use scenarios and it is misleading to even refer to them as things like short journey. The real time data in engine literature shows that often emissions are dominated by hard transient events.

It is helpful to understand results from laboratory studies by relating them to real-word measurements. In this study, agreement was found between engine tests and highway tunnel measurements for alkanes (C9-C13). Authors argue that "The emission factors in this study were comparable to on-road diesel vehicular emissions measured in Gentner et al. (2013), suggesting the results shown in this study are consistent with on-road diesel exhaust emissions." However, emission factors of hydrocarbons from diesel vehicles depend on many factors, such as vehicle type, driving condition, fuel type and aftertreatment devices. For example, Dallmann et al. (2012, EST) show that emission factors of pollutants for diesel vehicles span a wide range at the Caldecott tunnel where measurements in Gentner et al. (2013) were made. In the absence of further constraints, the comparison between measurements in this study and those

in Gentner et al. (2013) might not lead to this claim of consistency in both measurements. If the author's claim is true, it means that other factors, including the fuel type, vehicle type and emission control device, are insignificant for diesel emissions. This is in contrast to results that both chassis dynamometer testing and field measurements that catalyzed diesel particulate filters remove hydrocarbons very efficiently (May et al., 2014, AE; Dallmann et al., 2012 EST). Finally, this comparison is made with a 2010 fleet of on-road diesel trucks/vehicles. Shall we expect much lower emission factors from diesel vehicles from a present on-road fleet of diesel vehicles?

Authors mention that a typical DOC is expected to remove 50 to 70% of the total hydrocarbon emissions. The DOC tested in this study likely has much lower removal efficiency for hydrocarbons, much lower than 46% for measured VOCs if considering the fall substantially lower or no removal efficiency for IVOCs by this DOC (e.g. their data suggest less efficient for removal of C12 branch aliphatics compared to other VOCs with lower carbon number. This indicates that the DOC has no effect on IVOCs, predominated by species with carbon number >12). It seems likely that the DOC tested in this study does not represent the performance of most DOCs and one must be careful trying to generalize. Manufacturers carefully consider thermal management and other operating conditions to ensure that a DOC operates effectively. It was not clear that installation of the DOC on this engine took those factors into account.

---

## Author Comment (AC1) · 4 Dec 2017

**Response to Referees**

The authors thank the referees for their comments. Please find our responses below:

**1. Referee 1 Comments:**

In Europe almost half of all new passenger cars are diesel vehicles. The large number of diesel vehicles means that their emissions are an important source of air pollution in urban environments. This study focuses on speciated VOCs, including 16 individual and 8 groups of compounds and effects of a home-retrofitted DOC on the mass emissions and chemical composition of these VOCs from an older diesel engine. VOCs contribute to less than half of the organics in diesel exhaust with these specific compounds contributing an even smaller fraction (e.g. classic paper Schauer et al. EST 1999 or more recent papers by Gentner PNAS 2013 or Zhao et al. EST 2015). The majority of organic emissions from diesel vehicles are IVOCs. This paper does not provide this important context.

IVOCs were investigated. The saturation concentration of the investigated compounds ranged from $10^5$ to $10^8 \, \mu g \, m^{-3}$, capturing both VOCs ($C^* => 10^6 \, \mu g \, m^{-3}$) and IVOCs ($C^* = 10^3 – 10^6 \, \mu g \, m^{-3}$). Based on the work shown in Zhao et al. 2015 (see below figure), we capture the most abundant volatility fraction of the IVOCs with online sampling (in contrast to Zhao et al. and Schauer et al.). We provide quantitative measurements of speciated VOCs and IVOCs, emission measurements, which supported by reviewer 2, are relatively rare. Contextualisation and discussion of the abundance of IVOCs in diesel exhaust emissions has been added into the manuscript, see page 3, line 30. The manuscript title has been changed to further highlight the study of IVOCs, reading 'The Effect of Varying Engine Conditions on Unregulated VOC-IVOC Diesel Exhaust Emissions'.

[Figure]

The major weakness of the paper is that there is no substantive connection between the emissions data and air quality. Instead the paper focus on engine operations and emissions (an interesting topic but it seems outside the scope of ACP). Such a connection seems important for publishing in an atmospheric science journal. Therefore this study seems poorly suited to Atmospheric Chemistry and Physics.

The potential impacts of our findings on air quality has been discussed in section 3.5, within the limitations of the study. We cannot provide a substantive connection between the emissions data and air quality from one diesel engine, as one engine cannot be generalised to the entire vehicular fleet. Instead, we provide detailed information of how and why emissions change with varying engine conditions, specifically the effect of engine combustion efficiency and DOC HC removal efficiency on the exhaust gas composition. This information (with further work) could lead to improved refinement of air-quality models, predicting the impact of traffic emissions on air quality as a function engine temperature, which based on the result shown in this study, has a considerable effect on the composition

and abundance of the exhaust gas emissions. We have added further information on the possible atmospheric implications of our findings into the manuscript.

The paper makes numerous claims about the novelty and importance of the work. Many papers have examined VOC speciation of diesel exhaust and robust VOC speciation profile exist for diesel exhaust (the major problem with these profiles is the lack of IVOC data). The paper provides a very limited review of this literature and some readers may be confused on state of knowledge of diesel VOC emissions after reading the intro.

IVOCs have been investigated, see above comments. The point about a literature review of IVOC diesel profiling is addressed below.

The paper states multiple times that few studies have reported speciated emissions as a function of engine conditions. It is true that less is known about speciation as function of engine load and DOC then cycle based emissions (but much more work has been published than cited by this paper; a super quick search revealed multiple papers including Combust Flame, 118, 179, 1999; Atmos Env 42, 769, 2008, etc.). This paper, similar to the previously published work, clearly shows variations with engine loads and control technologies. The general trends (e.g. higher emissions at lower loads and changes in composition with loads) are consistent with the published literature. Similarly the results for the effectiveness of the DOC are similar to other studies (though at the low end of effectiveness presumably due to the retrofit nature of this application). More data are always good but the results are not especially novel from an emissions perspective. What are the implications of this new data from an atmospheric perspective (the focus of ACP)?

There is a considerable amount of papers which have investigated diesel exhaust emissions. Whilst every effort has been made to ensure we include at least the majority of these papers in the manuscript, there will be papers that we do not find, primarily because of the vast differences in titles and work aims. The papers cited above have now been included in the manuscript. See the below comments in response to the atmospheric implications of our work.

Air quality impacts depend on the integrated emissions (from many engines operated over a wide range of load conditions). The purpose of test cycles is to measure representative emissions that are relevant to atmosphere (emissions models like MOVES are moving to a more dynamic representation). Is there some problem with existing diesel VOC emissions profiles used by models and inventories that this paper is addressing? That was not clear (nothing jumps out to me), but I don't see an atmospheric question this paper is addressing. It seems more like engine / control technologies related questions. Maybe the paper belong in a different journal?

Air quality impacts do not just depend on integrated emissions using test cycles. Test cycles aim to represent on-road driving conditions. Whilst they cover a vast number of conditions in a single experiment, that all contribute to the emissions across the cycle, they cannot identify specific conditions that disproportionately contribute, nor can they isolate and completely characterise individual conditions. Our studies allow detailed characterisation of an instantaneous 'snapshot' and identify trends in these characteristics with changing engine conditions. The purpose of this work was not to repeat common exhaust measurements using driving cycles, but to ask the atmospherically relevant question, what engine factors are controlling the abundance and composition of VOCs and IVOCs in diesel exhaust emissions? This information could lead to the better constraint of atmospheric models and inventories. Using traffic flow models to identify vehicle use leading to engine conditions that are particularly polluting is not possible from averaged test-cycle derived emission factors. Furthermore, rather than using multiple test cycles to predict atmospheric emissions, we could use compositional trends based on one simple parameter (*i.e.* engine temperature) to estimate emissions in different driving conditions. The main results from this work show, (i) that engine temperature has a considerable effect on the composition and abundance of VOCs and IVOCs in the exhaust gas emissions, (ii) increasing combustion efficiency results in the increased formation of smaller, higher volatility *n*-alkanes in the

exhaust gas, which may be important in urban environments; *n*-alkanes are more efficient at producing SOA than their branched counterparts, and (iii) liquid fuel based estimates of SOA yields may be inconsistent with exhaust SOA yields, particularly at higher engine temperatures (*i.e.* high loads and speeds), due to the increased formation of smaller, more volatile *n*-alkanes at high engine combustion efficiencies (the contribution of *n*-alkanes increased by a factor of 1.6 in comparison to the contribution observed in cold-idle conditions (most similar to unburnt fuel). In addition to the above, IVOC measurements from diesel exhaust emissions are relatively rare, more measurements of these species are required (Gentner et al. 2017). We don't believe this manuscript in outside the remit of ACP, due to relevance of ACP literature to our work and the journal focus areas, laboratory measurements and chemical composition. We have also included further information on the possible atmospheric implications of our findings within the limitations of this study.

The study has used an advanced instrument (2D-GC-MS). However, the much of the analysis focuses on emissions of a commonly characterized subset of VOCs (n-alkanes and aromatics) that contribute a minor fraction of emissions. In addition, they are not the dominant source of SOA (e.g. dominated by IVOCs) or toxicity (carbonyls) from diesels. Not leveraging more the advanced 2D GC data seemed like a potentially missed opportunity.

We have investigated *n*-alkanes and branched aliphatics. The investigated compounds include, aliphatics (straight-chain and branched) and single-ring aromatics with two and three carbon substituents. These measurements were made using GC×GC-FID, not MS. Straight-chain and branched aliphatics constitute a considerable proportion of diesel exhaust emissions, primarily from unburnt fuel. Straight-chain and branched aliphatics have been found to constitute ~70% of the IVOC fractional composition at idle engine conditions (Cross et al. 2015). Contrary to the reviewer's assertions, we have investigated IVOC emissions, which we agree will contribute substantially to the SOA.

The engine is operated over extremely simple test cycles (e.g. constant speed and load or idle followed by constant load for a number of minutes). This is very unrepresentative of essentially all actual in-use scenarios and it is misleading to even refer to them as things like short journey. The real time data in engine literature shows that often emissions are dominated by hard transient events.

The reviewer is clearly missing the point of our study which was to gain a greater insight into the engine conditions controlling the composition and abundance of VOC-IVOCs in the exhaust emissions; experiments which cannot be performed using transient conditions due to averaging of emissions over entire driving cycles. However, we agree that use of the term 'short journey' could be misinterpreted. We have changed the term 'short journey' to 'cold loaded'.

It is helpful to understand results from laboratory studies by relating them to real-word measurements. In this study, agreement was found between engine tests and highway tunnel measurements for alkanes (C9-C13). Authors argue that "The emission factors in this study were comparable to on-road diesel vehicular emissions measured in Gentner et al. (2013), suggesting the results shown in this study are consistent with on-road diesel exhaust emissions." However, emission factors of hydrocarbons from diesel vehicles depend on many factors, such as vehicle type, driving condition, fuel type and aftertreatment devices. For example, Dallmann et al. (2012, EST) show that emission factors of pollutants for diesel vehicles span a wide range at the Caldecott tunnel where measurements in Gentner et al. (2013) were made. In the absence of further constraints, the comparison between measurements in this study and those in Gentner et al. (2013) might not lead to this claim of consistency in both measurements. If the author's claim is true, it means that other factors, including the fuel type, vehicle type and emission control device, are insignificant for diesel emissions. This is in contrast to results that both chassis dynamometer testing and field measurements that catalysed diesel particulate filters remove hydrocarbons very efficiently (Mayetal., 2014, AE; Dallmann et al., 2012 EST). Finally, this comparison is made with a 2010 fleet of on-road diesel trucks/vehicles. Shall we expect much lower emission factors from diesel vehicles from a present on-road fleet of diesel vehicles?

The Gentner et al. comparison has been removed from the manuscript.

Authors mention that a typical DOC is expected to remove 50 to 70% of the total hydrocarbon emissions. The DOC tested in this study likely has much lower removal efficiency for hydrocarbons, much lower than 46% for measured VOCs if considering the fall substantially lower or no removal efficiency for IVOCs by this DOC (e.g. their data suggest less efficient for removal of C12 branch aliphatics compared to other VOCs with lower carbon number. This indicates that the DOC has no effect on IVOCs, predominated by species with carbon number >12). It seems likely that the DOC tested in this study does not represent the performance of most DOCs and one must be careful trying to generalize. Manufacturers carefully consider thermal management and other operating conditions to ensure that a DOC operates effectively. It was not clear that installation of the DOC on this engine took those factors into account.

The results from the investigated DOC have not been generalised. We only discuss the hydrocarbon removal efficiency of the investigated DOC and whether the observed removal efficiency is in the expected range for a typical DOC. The life expectancy of a catalytic converter is ~ 50,000 miles. The average age of passenger cars in Europe is 10.7 years old (ACEA, 2017), with an average annual mileage of 13,000 km per annum (8078 miles). Thus, in Europe, the catalytic converter is likely to be changed at least once (on average) during the cars lifetime. The retrofitted nature of the DOC in this study, is therefore relevant to on-road vehicles and the age of the engine investigated (see page 5, line 24. Furthermore, the retrofitted DOC is the same as the VW catalyst that would be fitted to the engine if supplied for on-road use.

**2. Referee 2 Comments:**

General Comments:

This manuscript presents diesel exhaust measurements of from a light-duty diesel engine using an engine dynamometer. The study investigated the effects of engine load and use of diesel oxidative catalyst (DOC) on emissions of volatile and intermediate volatility organic compounds (VOCs/IVOCs) in diesel exhaust. The exhaust emissions were diluted in an atmospheric chamber and online sampling was conducted to measure speciated VOCs and grouped VOCs using a TD-GCxGC-FID system. The study results showed that VOC/IVOC emission rates and VOC profiles were greatly impacted by engine load and DOC efficiency. Intermediate volatility organic compounds (IVOCs) are quite difficult to sample and quantify accurately, and speciated IVOC emissions measurements are rare, particularly online measurements. Therefore, the publication of results from this work will be of significant benefit to the emissions research community to improve our understanding of IVOC emissions from a diesel engine and how they are impacted by engine conditions. However, there are several important issues raised below that must be addressed before this work is accepted for publication in ACP.

One major issue is that the rationale behind selection of the engine, fuel, experimental conditions was not articulated in the manuscript. Why was the particular engine with retrofitted DOC selected, and why was the goal to mimic Euro 4 emissions standard? I would imagine this engine/technology must be outdated compared to current and near future European light-duty vehicle fleet adhering to Euro 5 and 6 standards. Therefore, are these emission rate results at all relevant to the current European vehicle fleet?

The rationale behind the selection of the engine, emission control devices and experimental design have been added into the manuscript, see section '2.2 engine and exhaust sampling system' and newly added, '2.3 experimental design'. The auto-equivalent version of the engine has been used in several VW polo and Jetta models in the early 2000's and was chosen as an example of light-duty diesel engine. The aftertreatment was selected to meet Euro 4 emission control regulations required for such models. The retrofitted DOC is the same as the VW catalyst that would be fitted to the engine if supplied for on-road use. Euro 4 emission control regulations were first implemented for all new vehicles from

approximately January 2006, with Euro 5 emission control regulations starting post January 2011. Approximately 20% of the current EU diesel fleet are Euro 4 compliant (ACEA, 2017). Whilst vehicles conforming to Euro 5 and Euro 6 emission control regulations are increasing, the emission rates shown in this work are still relevant to the current EU diesel fleet.

Why were the specific experimental conditions chosen: specific engine loads, dilution ratios, speeds, multiple fuel batches? Based on the experiments listed on Table 1, the research strategy behind these measurements is very difficult to tease out. For example, in Section 3.1 the effects of engine load were discussed. However, because the DOC was also included, the results reflected the coupled effects of engine load and DOC that were interdependent. This is not ideal for a mechanistic study such as this work. Was it intended to study the effects of both simultaneously or was this an unintended consequence of the experimental design? Either way, no explanation was given. The rationale behind the experimental design and study conditions needs to be clearly described in order for the reader to interpret the results.

The rationale behind the experimental design has now been included in the manuscript. The rationale behind the engine loads, speeds, dilution ratios and fuel batches are also discussed below in the specific comments. The experiments discussed in this work formed a part of a wider project, 'combustion particles in the atmosphere, (Com-Part)'. This project focused on the systematic characterisation of the chemical and physical transformations of primary and secondary particles emitted from a light-duty diesel engine, under a range of atmospheric dilution and oxidation conditions. Measurements of the direct exhaust gas emissions were essential to investigate the chemical and physical transformations of the particles (discussed in a separate publication), but were not the focus of the project and hence why the experimental rationale (without explanation) can be difficult to tease out. It was intentional to study the combined effect of different engine conditions (*i.e.* load and driving scenarios) and the DOC on the exhaust composition, as these engine conditions are most representative of on-road diesel vehicles. In addition, as observed in the manuscript, the exhaust gas composition was observed to change due to the combined effect of engine combustion and DOC HC removal, rather than when just one effect (*i.e.* engine combustion efficiency) was present (see section 3.4 driving scenarios for further information and the specific comments below); highlighting the importance of studying the combined effect of the DOC and specific engine conditions on the exhaust gas composition.

The second major issue is that a large portion of the discussion in the supplementary information is extremely important and should be provided in the main text. I believe some discussion points brought up in the SI are actually more scientifically relevant and impactful than some of the extended discussion about engine load and DOC effects in the main text that have been previously studied by others.

The experimental reproducibility and discussion of the fuel batches have been removed from the SI and added into the main text of the manuscript, see sections 2.5 liquid fuel analysis, 3,1 experimental reproducibility and 3.1.1. ULSD fuel: Batch A and B.

Experimental reproducibility is vital experimental information; this information (e.g. measurement replicates, test replicates) needs to be clearly detailed in the main text in order for the reader to assess the data quality and statistical power of the results. In Table 1, which experiments were meant to be replicate tests, and are the numbers in parentheses the measurement replicates or something else? Also, the origins of the error bars and emission rate standard deviation values need to be explained in the main text. If they are from the calibration propagation of error uncertainty calculations in SI Section 3.1, then how were measurement and test replicates included in the errors?

The replicate experiments are now clearly highlighted in Table 1 (see specific comments below). The numbers in the parentheses in Table 1 are for reference, to aid experimental discussion in the main text. An additional table has also been added into the SI, showing the number of exhaust measurements in each experiment, sampling times and exhaust injection times, see SI, Table S2. A brief explanation of

the uncertainty in measured emission rates has been provided in the main text (see 3.1 experimental reproducibility) – the detailed discussion on the calculation of the uncertainties has been reserved for the SI; we do not believe this information should be in the main text. The emission rates from sample replicates have been averaged and are not included in the propagation of error. The majority of experiments had two replicate measurements (see SI Table S2) of the exhaust gas due to the long GC×GC-FID sampling time; an insufficient number of replicates to calculate a RSD required for propagation. The averaging of emission rates has now been included in the manuscript, along with a description in each figure of what the error bars represent.

The discussion in the SI indicated that dilution ratio had a substantial impact on IVOC emissions between two similar experiments. Given the wide range of dilution ratios used in this study, this issue warrants further discussion in the main text, particularly how this issue may have influenced the results. Furthermore, a discussion of the observed differences in emission rates for the two different fuel batches and linking them to fuel composition by 2DGC-TOF fuel analyses was very interesting and should be included in the main text.

This information has been included in the main text of the manuscript, see above comments and specific comments below.

The third major issue is that in Section 3.4 the comparison between these engine dynamometer emission rates with California tunnel measurements is misleading and not scientifically relevant. While I understand the desire to link these results to real world measurements, I don't see the scientific merit or rationale for making this particular comparison. This study used an engine, aftertreatment technology, and fuel that are not at all relevant for these CA on-road measurements. What I believe would be of much greater scientific benefit to the mobile source emissions research community is a thorough literature comparison of speciated IVOC measurements that are so rarely measured for at least light-duty diesel engines and vehicles from engine and chassis dynamometer studies. Are these measurements consistent with other light-duty (and perhaps also medium/heavy duty) diesel vehicles and how to they compare with Euro5 and Euro6 compliant engines/vehicles/fuels? If the IVOC data is lacking for modern vehicles with newer emission standard, how are the emissions expected to change from Euro 4 to Euro 5 and 6. Finally, how will the engine operation and other effects on VOC/IVOC emissions impact SOA formation from vehicle exhaust?

The Gentner comparison has been removed from the manuscript. A literature comparison was initially included but was later removed due to the lack of comparable studies. The vast majority of studies have reported emission rates in units of VOC mass emitted per distance travelled, units which are not comparable with this work. Furthermore, the large variation in the types of compounds speciated and their volatility, different engine types and emission control devices, made literature comparison extremely difficult. Whilst we agree that a literature comparison is extremely useful, there are too many important differences in the few studies we can compare with. We have instead, discussed the futility of the comparison of emission rates. It is worth noting, that (to our knowledge) there are no on-road light-duty diesel engine studies which report emissions rates with units comparable to this work. Furthermore, the lack of a distance measurement, means we are also unable to compare measured emission rates with set regulations. We agree that the comparison of IVOC emissions between Euro 4/ 5 and 6 would be a very useful piece of information. However, we cannot speculate as to how the speciated profile would change as there is insufficient data at present. These studies are still in their infancy. The nomenclature 'speciated' is unfortunately rather vague. This has led to it being used in different ways and to describe different fractions of IVOCs in the literature. The inter-group variability of 'speciated emissions' is large, meaning it is impossible at the minute to attempt the kind of analysis requested.

**3. Referee 2 Specific Comments and Author Responses:**

Section 1. Introduction: there are some repetitive statements (Page 2, Lines 17-20; Page 3, Lines 34-36; Page 4, Lines 1-2, 5-7). The introduction could be made more concise by removing the repetitive text.

Repetitive text removed.

Page 4, Lines 10-12. The introduction referenced research that has previously studied the effects of DOC and engine operating parameters on vehicle exhaust composition, e.g. Chin et al. 2012. Other studies have also investigated these effects on diesel vehicles/engines (e.g. Ballesteros et al. (2014), Zhu et al. (2013)). Can this statement be further clarified so that it is more accurate?

The above studies have been added into the manuscript and the text further clarified.

Page 4, Line 28. Please provide hygrometer vendor information.

Information added to manuscript.

Section 2.2. Section title is misleading. This section not only describes the engine but also exhaust sampling system. Please provide more detailed information on the engine specifications and about DOC usage status (mileage or rapid thermal aging hours) as this can significantly impact the emissions. Provide rationale behind the selection of the specific engine, aftertreatment and why Euro 4 emission control was approximated. Please provide a schematic of the dynamometer and sampling system (this may be included in the SI). Please describe all engine operating conditions including driving scenarios and include rationale behind the experimental design.

Section title has been changed to 'Engine and exhaust sampling system'.

The DOC was newly-bought (0 mileage hours). Page 5, line 21, now reads; 'The engine was mounted on an eddy current dynamometer rig (CM12, Armfield Ltd, Hampshire, UK) and the exhaust connected to a new (0 mileage hours) retrofitted DOC.'

The rationale behind the selection of the engine has been added into the manuscript, see page 5, lines 24 - 26. The engine does conform to Euro 4 emission control regulations; the use of 'approximated' has been removed. A schematic of the dynamometer and sampling system has been added to the SI. A more detailed description of the engine operating conditions and rationale behind the experimental design, has been added in to a new section, '2.3 experimental design', starting on page 6, line 10).

Page 5, Line 12-13. Was the MAC filled before exhaust was introduced? How were the dilution ratios decided upon and why did they vary from experiment to experiment? How long was the diluted exhaust allowed to equilibrate before sampling took place and were there any apparent losses in IVOCs over time?

The MAC was filled with clean air prior to the introduction of the exhaust emissions. Exhaust dilution ratios were varied to represent a range of ambient conditions from near to downwind of an emission source, capturing the chemical and physical transformations of semi-volatiles in the exhaust emissions with varying dilutions. The above has been added into the manuscript, see page 6, lines 21 -24.

The GC×GC-FID started sampling during chamber cleaning and sampled for 26 minutes per analysis. The sampling start time was on average 13 minutes after injection. No apparent losses of the IVOCs were observed during sampling. The average RSD of the investigated compounds from replicate exhaust measurements, over the longest measurement period (~ 2 hours), was 6.4%. This information has been added into the manuscript, see page 8, lines 18 – 24.

Page 5, Line 18. Please explain why two batches of diesel fuel were used. Please include any fuel composition or fuel property analyses that were conducted. What was the sulfur content of the two batches? Do both batches of the diesel fuel meet Euro 4 or Euro 5 specifications? Please discuss differences in fuel composition here. What is meant by "standard"?

The two batches of fuel were of same specification and obtained from the same local fueling station. A second batch of fuel was required due to a considerable increase in the number of planned experiments. The fuel used was 'standard' European ULSD and is Euro 5 compliant, conforming to EN590 specifications. The sulphur content was < 10 ppm. Readers have been referred to the 2009 EC directive for further information on the fuel specifications. 'Standard' refers to standard European specifications. The above has been added to the manuscript, see page 6, lines 3 – 9.

Page 5, Line 25. What temp was the line heated to?

The line was heated to ~ 70 °C. Page 7, line 15 now reads, 'The stainless steel tubing was heated to ~ 70 °C to reduce condensational losses of VOCs.'

Page 6, Line 17. How often was the calibration checked by a standard? Were MDLs determined? If so, explain how and provide MDL values for all compounds. This can be provided in the SI.

Calibrations were performed weekly using the NPL gas standard, or more frequently during instrument maintenance periods. The MDLs for the investigated compounds using this instrument, can be found in the SI of Dunmore et al. (2015). The above has been added into the manuscript, see page 8, lines 7 – 9.

Page 7, Line 1-4. The engine operating conditions should be described in detail in the experimental as well as justification for why these specific conditions were chosen.

This has been added in the manuscript under section '2.3 Experimental design', see above comment.

Page 7, Line 9. Not enough evidence has been provided to identify the grouped VOCs "branched aliphatics", particularly because these were not GC/MS measurements. These would be better labelled as unspeciated or unresolved organics. The grouped VOCs should also not be called speciated VOCs because they are not individually speciated. Therefore, sum of SpVOC is a misleading term and should also be changed to minimize confusion. Also, please discuss why this narrow range of VOCs was studied? Was it intentional to focus on IVOCs or was this due to sampling/instrumental limitations?

Branched aliphatics can be identified using GC×GC in combination with commercially available standards. Two different stationary phases were used, separating compounds by two physical properties, boiling point (see Figure 1, x-axis, increasing from left-to-right) and polarity (y-axis, increasing from bottom-to-top). This separation creates a characteristic space where compounds are grouped by similar physical properties, providing the 'aromatic' and 'aliphatic' bandings observed with this technique (*e.g.* Hamilton and Lewis (2003),(Dunmore et al., 2015); bandings added to Figure 1 for clarity). Straight-chain aliphatics have lower boiling points than their branched counterparts with the equivalent number of carbon atoms. For example, *n*-nonane has a lower boiling point than 2-methyloctane ($C_9$ branched aliphatic). Thus, branched aliphatics elute to the righthand side of the equivalent carbon number straight-chain aliphatic in the chromatogram (see Figure 1). The use of external standards to identify the straight-chain aliphatics, in-turn allows the identification of the branched aliphatics based on their boiling point (*i.e.* chromatographic location) (*c.f.* Dunmore et al. 2015)

Whilst the branched aliphatics have not been individually speciated, 'unspeciated' or 'unresolved organics' do not accurately describe these groupings. The branched aliphatics have identified functionality and have been speciated by the number of carbon atoms within each group, retaining some level of speciation; unlike the unidentified complex mixture (UCM) often observed with the use of one-dimensional GC. Subsequently, the term SpVOC has not been removed from the manuscript.

Diesel exhaust emissions consist mainly of aliphatic and aromatic species with a carbon range of ~ $C_5$ to $C_{22}$. The measured compounds include aliphatics and single-ring aromatics with a carbon range of $C_6$ to $C_{13}$. This narrow carbon range is the result of instrument temperature constraints. The boiling points of compounds above ~ $C_{15}$ are too high to be removed from the column at the maximum operating temperatures of the modulator, resulting in these species not being detected. IVOCs were intentionally

studied, with the aim of characterising the largest volatility range possible within instrument limitations. This has been added into the manuscript, starting page 18, line 9.

Page 7, Line 11. How often were the retention time windows confirmed? Were any internal standards used?

The shift in retention time that this comment relates to, occurred as a result of instrument maintenance, between the first and second set of experiments. As mentioned above, calibrations and thus retention times were confirmed weekly using the NPL gas standard, or more frequently during instrument maintenance periods.

Internal standards ideally need to be of similar chemical speciation to the sample. The composition of the exhaust emissions was largely unknown and thus it is likely the use of internal standard may add additional mass to compounds already present in the exhaust gas, resulting in sample contamination (*i.e.* unable to distinguish sample and standard concentrations).

Page 7, Line 23-24. Both reproducibility and dilution effects are important enough to merit detailed discussion in the main text. Please see general comments above.

The discussion reproducibility and dilution effects have been added into the main text, see section '3.1 experimental reproducibility'.

Page 7, Line 29-30. While similar sets of conditions have been compared in the discussion, emission factors are all presented for every experiment, allowing the reader to freely compare between experiments without fully understanding the impacts of differing dilution ratios unless they read the SI in detail. This is why a full discussion of the effects of dilution ratios is needed in the main text.

See above comment.

Section 3.1 Why were these loads chosen? Please state clearly whether these experiments (and for each subsection in the Results section and in all figure captions) are from tests with or without DOC? This section title is also misleading because the combined effects of engine load and DOC were studied simultaneously.

The engine loads investigated ranged from 0 (no load) to 53% (maximum engine load which could be safely applied by the dynamometer in the experimental setup). Within this range, 0, 30, 40 and 53% load were investigated. The experiments shown in this work form only a subset of total number of experiments performed, as a part of the wider project, combustion particles in the atmosphere. The GC×GC-FID was not in operation during every experiment. The above has been added into section 2.3 experimental design, starting page 6, line 10.

There appears be too much emphasis on the DOC. Only one experiment was performed without a DOC to investigate the hydrocarbon removal efficiency of the catalyst and this is clearly shown in Table 1. A separate sub-heading has also been used (3.3. DOC removal efficiency) to separate these results from the other text. The other results sections 'engine load' and 'driving scenarios' show the combined effect of the investigated parameters with the DOC; conditions most representative of on-road vehicles. We do not believe these section titles are misleading. However, to avoid confusion, we have clarified the text in section 2.3 Experimental design (page 7, line 2), stating; 'All experiments except experiment 3 (see Table 1) were performed with the DOC.'

Page 8, Line 7. Is this statement suggesting that engine load is the most important factor that controls VOC emission rates, whereas other factors, e.g. aftertreatment and fuel, are less important? If not, please clarify. Also, please state clearly that the assumption was made that engine load is linearly related to engine combustion efficiency or provide ancillary measurements to confirm combustion efficiencies for each experiment.

We don't mean to suggest that engine load is the 'most important factor' that controls VOC emission rates. Rather, that a pattern is observed in the literature (and in this work) between increasing engine load and decreasing VOC emission rates. This same pattern is observed with the use of different fuel compositions and emission control devices, when all conditions except engine load are kept consistent in each study. The statement has been shortened to avoid confusion, reading (page 11, 20 -23), 'This trend of decreasing VOC emission rates with increasing engine load has been observed in a number of previous studies for light-duty and medium-duty diesel vehicles and can be explained by considering the engine operation.'

This linearity comment relates to page 11, lines 15 – 18 in the original manuscript (to avoid confusion; driving scenarios discussion), Linearity is not assumed in the engine load section. The relationship between engine load (and thus engine temperature) is relatively linear with engine combustion efficiency. Engine temperature increases with increasing engine load. The engine requires more power to achieve higher loads, which is achieved by increasing the amount of fuel injected into the combustion chamber. The increase in fuel, increases fuel combustion, generating more power and increasing temperature (combustion generates heat) (*e.g.* Kumar, 2009). This increase in temperature, in-turn increases engine combustion efficiency, as fuel components are more readily combusted at higher temperatures (Haywood (1988)). The relationship between engine temperature and combustion efficiency can be observed in Mikalsen and Roskilly (2008); shown as percentage in-cylinder heat transfer losses *vs.* engine efficiency, although as heat transfer losses are minimised, temperature increases and therefore supports the above. It is worth noting however, at high engine loads and relatively low speeds (not used in the driving scenario experiments), engine temperature and combustion efficiency eventually plateau due to a too lean air/fuel ratio, resulting in incomplete combustion (see Haywood (1988) for further information). At a constant speed, the air flow into the combustion chamber remains constant, but the amount of fuel injected continues to increase with increasing engine load. This results in insufficient oxygen content to burn all the fuel (*i.e.* incomplete combustion) and thus temperature and engine combustion efficiency plateaus. A more concise version of the above has been added into the manuscript, see page 15, line 32.

Page 8, Line 18, 20. Should the reference be Chin et al. (2012)?

Yes. The reference year has been changed.

Page 9, Lines 16-22. This text contradicts what is shown in Figure 2a, 2b and previous discussion on Page 8.

There are two factors influencing VOC emissions here, the DOC hydrocarbon removal efficiency and engine combustion efficiency. From 0 to 40% engine load, the compositional profiles display the effect of both increasing DOC hydrocarbon removal efficiency and engine combustion efficiency. The DOC hydrocarbon removal efficiency is near maximum at 40% engine load and thus cannot account for the considerable compositional shift observed at 53% engine load (see Figure 2B). This considerable increase in the percentage contribution of the $C_7$ to $C_{13}$ *n*-alkanes to the $\sum$SpVOC emissions at 53% engine load, has been attributed to engine combustion. We suggest that this high engine combustion efficiency results in an increase in the abundance of the *n*-alkanes to the $\sum$SpVOC emissions. This increase in the percentage contribution of the *n*-alkanes is not observed from 0 to 40 % engine load (appearing contradictory) and is likely the result of the DOC masking the effect of engine combustion efficiency on the exhaust composition. We later show this in the driving scenario experiments, where the effect of engine combustion efficiency on the VOC emissions can be observed without the DOC interference (DOC hydrocarbon removal efficiency is negligible in all driving scenario experiments). In the driving scenario experiments, we observe that as the internal combustion efficiency increases the contribution of the *n*-alkanes to the $\sum$SpVOC emissions also increases.

Page 13, lines 19 and 20 have been removed to avoid confusion and the text further clarified from lines 26 to 30.

Page 10, Lines 16-17. Did Alam et al. use the same type of DOC or was it different from what was used in this study? It may also be interesting to compare the results of Liu et al. (2010) with this work.

Alam et al. used a similar DOC (mixed platinum and rhodium). This has been added into the manuscript.

The results of Liu et al. (2010) and (2008) are very interesting, although we did not feel that this work was directly applicable in the DOC removal efficiency section.

Page 10, Lines 26-30. Please show how the conditions of these three scenarios change in a diagram in the experimental section or SI.

The cold-start experiment includes only one applied engine load and speed which can be observed in Table 1. A diagram for the engine sequences ('short journey' and 'warm following load') however, have been added to SI, Figure S2.

Page 11, Line 17-18. Relative abundances discussed here?

The relative abundance discussion starts at line 16 in the original manuscript.

Page 11, Line 18-22. It is not clear why the effect of the DOC are brought up here. It does not logically fit with the rest of the discussion in this paragraph.

See above comment to page 9, lines 16-22. The comments here refer to the additional information uncovered, the result of being able to observe the effect of only combustion efficiency on the exhaust emissions. The text has been further clarified to make this text easier to follow/understand. See page 16, lines 3-7.

Section 3.4. Please see general comments on this section. It would be extremely useful to conduct a literature comparison to assess whether the emission factors from this study are consistent with similar studies. The comparison with Gentner et al. (2013) is misleading and should not be included.

The Gentner et al. comparison has been removed from the manuscript. A literature comparison could not be performed. We have instead discussed the futility of the comparison of emission rates (see general comments above). See starting page 16, line 33.

Page 12. Line 32-33. Aldehydes are not at all mentioned here and represent arguably the most important air toxic/hazardous VOCs emitted from modern diesel vehicles.

Aldehydes are not mentioned in the manuscript as these compounds were not measured. We have not discussed other toxic/carcinogenic species that we have measured in manuscript and thus the addition of aldehydes, whilst we agree are important, does not fit within the discussion. Also, aldehyde measurements from diesel exhaust are fairly common.

Page 13. Lines 3-7. Please discuss differences between Euro 5 and Euro 6 emission standards in relation to the approximated Euro 4 emission standards simulated here and how modern vehicles and their aftertreatment technologies are specifically changing. How are these changes expected to impact VOC emissions? Emission control technologies targeting other pollutants may still have significant effects on VOC emissions. This is where the literature comparison of emission factors from this study with emission studies with Euro 5 and Euro 6 vehicles would provide insight (see general comments).

See general comments above.

Page 13. Lines 7-8. Please see general comments about this comparison.

Text removed.

Page 13. Lines 11-13. Please see earlier comment on the veracity of this statement.

We do not believe this comment requires modification. Several compositional changes observed in the exhaust gas with different engine conditions agreed with the results of previous studies. The earlier comment ('Page 8, Line 7') has been changed.

Table 1. Please clearly indicate which tests were replicate tests and which tests had replicate measurements. Figure 1. What are the compounds below B and above G? What is the peak above #8? Why does benzene elute after toluene – are those labels correct? Have any attempts been made to quantify the lighter compounds and to assess total unspeciated organics?

Replicate experiments have been highlighted using superscript letters, see Table 1. The compounds below B are likely single-ring aromatics with four carbon substitutions, based on the elution patterns in Dunmore et al. (2015). The compounds above G are possibly single-ring aromatics with five carbon substitutions, although this cannot be determined without commercially available standards. The labels for benzene and toluene are incorrect. Benzene elutes before toluene; the manuscript has been corrected. No attempts have been made to assess to quantify the light compounds.

Figure 2. What do error bars represent? Are these tests with or without DOC? Toluene and benzene colors are difficult to distinguish.

The error bars represent the uncertainty in the measured VOC emission rates based on; (i) the standard deviation in the replicate measurements of the calibration standard and the reported uncertainty in the standard VOC mixing ratios, (ii) standard deviation of the replicate measurements of the liquid standards used for the calculation of the RRF (where applicable), and (iii) a 5% standard deviation in the chamber volume. The uncertainty in the measured emissions are discussed in the SI, Section 1.1. A description of the error bars has now been included in all applicable Figures.

These tests are performed with a DOC. See section 3.1 comments above.

The benzene colour has been changed.

Figure 4. Evidence was not provided to speciate into aliphatic and aromatic components.

See above comments, 'Page 7, Line 9'.

Figure 5. See general comments on this comparison.

Figure removed. See section 3.4 comments above.

Supplementary Information Page 4, Lines 13-14. Why was toluene used and not an n-alkane?

Toluene, in comparison to the *n*-alkanes, was much more resolved and easier to distinguish in the chromatogram. The reference compound (*i.e.* toluene) does not need to be structurally similar to the analyte (*i.e. n*-alkane), to calculate the analyte concentration. The response of a FID is proportional to the number of carbon atoms in a compound. The structure of the compound has a very small effect (within detector variation) on the detector response. The resolution of the reference compound is much more important, as poor resolution will increase the measured area of the reference compound, in-turn inaccurately increasing the concentration of the analyte. A more concise version of the above has been added into the SI, see page 4, line 14.

References

ACEA Report: Vehicles in use Europe 2017. (URL:
http://www.acea.be/uploads/statistic_documents/ACEA_Report_Vehicles_in_use-Europe_2017.pdf)

Dunmore, R. E., Hopkins, J. R., Lidster, R. T., Lee, J. D., Evans, M. J., Rickard, A. R., Lewis, A. C., and Hamilton, J. F.: Diesel-related hydrocarbons can dominate gas phase reactive carbon in megacities, Atmos. Chem. Phys., 15, 9983-9996, 10.5194/acp-15-9983-2015, 2015.

Gentner, D. R., Jathar, S. H., Gordon, T. D., Bahreini, R., Day, D. A., El Haddad, I., Hayes, P. L., Pieber, S. M., Platt, S. M., de Gouw, J., Goldstein, A. H., Harley, R. A., Jimenez, J. L., Prévôt, A. S. H., and Robinson, A. L.: Review of Urban Secondary Organic Aerosol Formation from Gasoline and Diesel Motor Vehicle Emissions, Environmental science & technology, 51, 1074-1093, 10.1021/acs.est.6b04509, 2017

Hamilton, J. F., and Lewis, A. C.: Monoaromatic complexity in urban air and gasoline assessed using comprehensive GC and fast GC-TOF/MS, Atmospheric Environment, 37, 589-602, https://doi.org/10.1016/S1352-2310(02)00930-5, 2003.

Heywood J. B., Internal combustion engine fundamentals (London: McGraw-Hill Inc. 1988). ISBN 0-07-100499-8.

Mikalsen, R and Roskilly, A., The fuel efficiency and exhaust gas emissions of a low heat rejection free-piston diesel engine, SAGE, 233, 4, DOI: 10.1243/09576509JPE653, 2008.

---

## Author Response (AR3)

Dear Prof. Ulrich Pöschl,

5    Following the executive co-editor review of our previously submitted manuscript (acp-2017-603), please find enclosed a revised version entitled: "Technical Note: Use of an Atmospheric Simulation Chamber to Investigate the Effect of Different Engine Conditions on Unregulated VOC-IVOC Diesel Exhaust Emissions". As discussed, we would like to submit our manuscript as a technical note for publication in Atmospheric Chemistry and Physics. The original manuscript has been revised in-line with the executive co-editor instructions. The changes to the manuscript can be observed below (highlighted in red).

10    Please let me know if you would like us to make any further revisions.

Yours Sincerely,

Dr. K. Pereira and co-authors

[revised manuscript text omitted]

---

## Author Response (AR4)

Co-editor comments:

Dear Dr. Pereira and Colleagues,

Following up on our preceding exchange and for consistency with the journal scope of ACP (atmospheric science rather than environmental technology), please adjust the abstract and conclusions of the revised manuscript as suggested in the attached file or in a similar way, further emphasizing the atmospheric science approach applied and described in this study and technical note (use of atmospheric simulation chamber and measurement techniques) rather than particular aspects of combustion technology (efficiency of a single Euro 4 engine and DOC).

Many thanks and best regards,
Uli Pöschl

Author response:
Dear Prof. Uli Pöschl,

Thank you for your comments. We have revised the manuscript in accordance with the above comments and the attached document. Some of the revisions in the attached document were shown as small red arrows which were difficult to see. I believe we have addressed all the comments/revisions. However, please let us know if we have accidentally missed any changes.

*Attached document comment, page 1 line 26 "correct?"*
The values shown are correct. We show the total percentage removal efficiency of the investigated DOC for all measured compounds, followed by the compound class dependant removal efficiencies, *i.e.* 'aromatic' and 'aliphatic'. These values have been calculated by subtracting the emission rates 'without a DOC' from 'with a DOC' and consequently are independent of one another, therefore not totalling to 100%. Please see Table 2 and the below calculation for further clarification.

Removal efficiency $(mg\ kg^{-1})$ = emission rate without DOC $(mg\ kg^{-1})$ – emission rate with DOC $(mg\ kg^{-1})$
Removal efficiency (%) = (removal efficiency $(mg\ kg^{-1})$ / emission rate without DOC $(mg\ kg^{-1})$) × 100

To make the above clearer we have reworded page 1 line 26. Page 1 line 26 now reads, "The investigated DOC was found to remove 43 ± 10 % 
[revised manuscript text omitted]

---

## Author Response (AR5)

Dear Dr. Pereira and Colleagues,

Thanks and yes, a couple of editing suggestions were apparently missed. Please consider specifying the nature of the values reported in the abstract (arithmetic mean +/- standard deviation, correct?) and implement a final clarification of the match between manuscript contents and journal scope in the last sentence of the abstract as follows:

"To our knowledge, this is the first study using an atmospheric simulation chamber to separate the effects of the DOC and combustion efficiency on the exhaust gas composition."

Many thanks and best regards,
Uli Pöschl

Dear Prof. Uli Pöschl

Thank you for your comments. Please accept our apologies for the two comments which were missed from the previous revision. We have revised the manuscript as described above.

Yours sincerely,
Dr. Pereira

[revised manuscript text omitted]

---

## Author Response (AR6)

Co-editor comments:

Dear Dr. Pereira and Colleagues,

Thanks for the revised submission. It seems, however, that a couple of adjustments and clarifications are still missing. (1) Please define the term "experimental uncertainty" in the main text. It seems not appropriate to introduce such a term in the abstract without specifying it in the main text. (2) Please implement a final clarification of the match between the content/message of the technical note and the journal scope of ACP in the last sentence of the abstract as follows: "To our knowledge, this is the first study using an atmospheric simulation chamber to separate the effects of the DOC and combustion efficiency on the exhaust gas composition."

Many thanks and best regards,
Uli Pöschl

Author response:

Dear Prof. Uli Pöschl

Thank you for your comments. The experimental uncertainty has now been defined in the main text. Please see page 10 lines 1 to 5, page 11 line 3 and page 24 line 4. The final clarification between the journal scope and the technical note has been included in the abstract and now reads, "
[revised manuscript text omitted]